# FLAT REWARD IN POLICY PARAMETER SPACE IMPLIES ROBUST REINFORCEMENT LEARNING

**Hyun Kyu Lee[1], Sung Whan Yoon[1,2]\***
[1]Graduate School of Artificial Intelligence and [2]Department of Electrical Engineering
Ulsan National Institute of Science and Technology
{dnwldlwl, shyoon8}@unist.ac.kr

## ABSTRACT

Investigating flat minima on loss surfaces in parameter space is well-documented in the supervised learning context, highlighting its advantages for model generalization. However, limited attention has been paid to the reinforcement learning (RL) context, where the impact of flatter reward landscapes in policy parameter space remains largely unexplored. Beyond merely extrapolating from supervised learning, which suggests a link between flat reward landscapes and enhanced generalization, we aim to formally connect the flatness of the reward surface to the robustness of RL models. In policy models where a deep neural network determines actions, flatter reward landscapes in response to parameter perturbations lead to consistent rewards even when actions are perturbed. Moreover, robustness to action perturbations further enhances robustness against other variations, such as changes in state transition probabilities and reward functions. We extensively simulate various RL environments, confirming the consistent benefits of flatter reward landscapes in enhancing the robustness of RL under diverse conditions, including action selection, transition dynamics, and reward functions. The code for these experiments is available at https://github.com/HK-05/flatreward-RRL.

## 1 INTRODUCTION

Reinforcement Learning (RL) has emerged as a powerful strategy for optimizing sequential decision-making tasks, with applications ranging from robotics to game playing and beyond (Mnih et al., 2015; Silver et al., 2016). Despite its great success, one of the critical challenges in RL remains its susceptibility to variations in the environment, which can significantly degrade performance (Peng et al., 2018). This challenge has driven interest in Robust Reinforcement Learning (RRL), a subfield focused on enhancing the generalization and resilience of RL algorithms against such environmental variabilities (Morimoto & Doya, 2005b; Rajeswaran et al., 2017; Pattanaik et al., 2018). Traditional RRL approaches, such as robust policy optimization and action robust reinforcement learning, have made strides in this direction. However, these methods often involve complex implementations and limited algorithmic diversity, especially when considering the broader landscape of RL (Pinto et al., 2017; Tessler et al., 2019).

In contrast, recent advancements in supervised learning have highlighted the importance of the landscape of loss values in the model parameter space. Specifically, flatter minima—with smooth varying loss values around them—are preferred as they exhibit strong generalization capabilities and are less sensitive to perturbations in the input distribution (Keskar et al., 2017; Hochreiter & Schmidhuber, 1997; Dinh et al., 2017; Foret et al., 2021). This raises a natural question: *Does a flat reward landscape imply robustness in RL against environmental changes?*

While a few works have begun to explore this question, finding that optimization for flatter reward maxima sometimes shows meaningful gains against varying environments (Lee et al., 2024), and that RL methods with flatter reward surfaces tend to exhibit robust performance (Sullivan et al., 2022), these prior studies often rely on a naive transfer of benefits observed in supervised settings to RL optimization. Moreover, they primarily provide empirical evidence without establishing a

---

*: Corresponding Author

rigorous link between flat rewards and robustness. To the best of our knowledge, the potential of flat reward landscapes in RL remains largely unexplored.

To illustrate the practical significance of this concept, consider the following preliminary experiment. We ran a 2D navigation task where an agent must move from a fixed start position to a goal position, with a very narrow path between two obstacles (see Fig. 1). In this action-critical setting, slight action perturbations can severely deteriorate performance. We compare a standard Proximal Policy Optimization (PPO) algorithm with our variant of PPO enhanced with Sharpness-Aware Minimization (SAM), denoted as 'SAM+PPO,' which pursues flatter reward maxima.

As shown in Fig. 1, the paths generated by multiple trials are depicted in orange. The standard PPO agent tends to follow the shortest path, bringing it dangerously close to the obstacles. This behavior makes the agent susceptible to failure when actions are slightly perturbed, as minor deviations can result in collisions. In contrast, the SAM+PPO agent maintains a safer margin from the obstacles, avoiding the narrow path and demonstrating more robust decision-making in the presence of action perturbations. This preliminary experiment highlights how pursuing flatter reward maxima can lead to enhanced robustness in RL agents.

Figure 1: Trajectories of agents in the 2D navigation task: (a) PPO tends to take the shortest path, risking collision with obstacles; (b) SAM+PPO maintains a safer margin, demonstrating robustness to action perturbations.

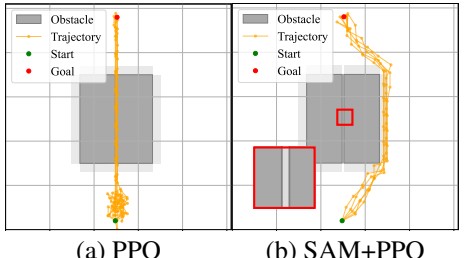

(a) PPO  (b) SAM+PPO

In this paper, we delve deeper into the concept of flat reward landscapes and their implications for robust reinforcement learning. Our main claim is that flatter reward maxima rigorously lead to robustness against action perturbations (**Proposition 1**). We also provide insights into how this robustness extends beyond action robustness to encompass transition probability and reward robustness. To validate our claims, we conduct extensive experiments in various MuJoCo environments (Todorov et al., 2012), including Hopper, Walker2d, and HalfCheetah, by varying actions, transition probabilities, and rewards.

Our work is outlined as follows: Section 2 reviews related prior works, and Section 3 presents the preliminaries with a focus on Robust Markov Decision Processes and SAM. In Section 4, we introduce our definition of flat reward maxima and establish the rigorous link between flat rewards and action robustness. Section 5 presents extensive simulations and analyses that validate the effect of flatness on the reward surface in various RL benchmarks. We further compare our flat reward policy with existing robust RL methods, emphasizing gains in performance and computational efficiency (see also **Appendix E**). Finally, Section 6 concludes the paper.

## 2 RELATED WORK

### 2.1 ROBUST REINFORCEMENT LEARNING

Robust Reinforcement learning (RRL) focuses on developing agents that maintain high performance despite uncertainties in the environment. Early approaches in this domain were predominantly **model-based**, leveraging explicit models of the environment to account for uncertainties. Morimoto & Doya (2005a) utilized $H_\infty$ control theory to formulate RL as a zero-sum game between the agent and nature, providing robustness through worst-case scenario optimization. Similarly, Iyengar (2005) and Nilim & El Ghaoui (2005) introduced Robust Markov Decision Processes (RMDPs) that model uncertainties in transition probabilities and reward functions, enabling the derivation of robust policies with theoretical guarantees. Several studies have proposed model-based robust RL algorithms in the tabular domain, with a strong emphasis on improving sample efficiency (Yang et al., 2022; Zhou et al., 2021; Panaganti & Kalathil, 2022).

However, model based methods often face scalability issues due to the computational complexity involved in high-dimensional spaces (Xu & Mannor, 2010; Wiesemann et al., 2013; Mannor et al., 2016; Goyal & Grand-Clement, 2023). To overcome these limitations, the focus gradually has been shifted towards **model-free** approaches, which do not require explicit environmental models, making it more scalable and applicable to complex tasks.

Recent research has explored model-free robust RL in settings with unknown uncertainty sets, where the agent pursues robustness without precise knowledge of the environmental uncertainties. Tessler et al. (2019) proposed Action Robust Reinforcement Learning, which models uncertainties in the action space and learns policies robust to adversarial action perturbations. Mankowitz et al. (2019) introduced Robust Reinforcement Learning via Regularized Policy Optimization, applying regularization techniques to improve robustness without explicit adversarial training. Derman et al. (2021) developed the TWICE algorithm, a model-free method that achieves robustness by accounting for both variance and bias in policy evaluation. Roy et al. (2017) introduced a method for robust Q-learning that accounts for uncertainties in transition dynamics by optimizing the worst-case performance over an ambiguity set, which is estimated online without prior knowledge (Wang & Zou, 2021; Badrinath & Kalathil, 2021).

Advancements in **policy-based** methods have further propelled robust RL (Kumar et al., 2024; Li et al., 2022; Grand-Clément & Kroer, 2021). Pinto et al. (2017) introduced Robust Adversarial Reinforcement Learning (RARL), where an adversary is trained alongside the agent to perturb the environment, enhancing the agent's robustness. By directly parameterizing and optimizing the policy, these methods offer advantages in handling continuous and high-dimensional action spaces. Wang & Zou (2022) developed a model-free robust RL algorithm that utilizes a worst-case policy gradient approach to handle unknown uncertainties in the environment. Li et al. (2022) proposed First-Order Constrained Optimization in Policy Space (FOCOPS), which ensures robustness by enforcing safety constraints during policy updates.

Our work aligns with this progression, focusing on model-free, policy-based RL to enhance robustness through the optimization of flat reward surfaces. To the best of our knowledge, the flat reward surface of policy-based RL has not yet been deeply explored in the view of robustness.

## 2.2 Flat Minima and Flatness in Reinforcement Learning

The concept of flat minima in the loss landscape has been extensively studied in supervised learning, where flatter regions are associated with better generalization and robustness to perturbations (Hochreiter & Schmidhuber, 1997; Keskar et al., 2017; Dinh et al., 2017; Dziugaite & Roy, 2017; Petzka et al., 2021; Andriushchenko et al., 2023). Several methods have been proposed to find flat minima, including Entropy-SGD (Chaudhari et al., 2019), Stochastic Weight Averaging (SWA) (Izmailov et al., 2018), and SWA with Densely (SWAD) (Cha et al., 2021).

Among the prior works, Foret et al. (2021) introduced Sharpness-Aware Minimization (SAM), which directly embeds the loss flatness into the objective function, enabling the search for minima with flatter surrounding regions. SAM has shown significant improvements in generalization performance across various supervised learning scenarios, including domain generalization (Cha et al., 2021) and federated learning (Qu et al., 2022; Caldarola et al., 2022; Sun et al., 2023).

Lee et al. (2024) applied SAM to policy gradient methods, observing empirical improvements in policy robustness. However, the prior work shows the following limitations: It primarily focuses on exhibiting partial empirical evidence without a formal bridge between the flatness in the reward landscape and the robustness in RL. Also, the effects of reward flatness are not carefully scrutinized in the multiple key perspectives of RL, i.e., action, transition probability, and reward.

Our work is also based on the SAM-based policy gradient methods, but we do not focus on further pursuing algorithmic advances. Our work distinguishes itself by providing a further understanding of how the flatter reward maxima contribute to the robustness of RL agents, which remains unknown yet. Also, we tested how flatness improves the robustness of RL against the perturbations of actions, transition probabilities, and rewards.

## 2.3 Reward surface visualization

Ilyas et al. (2020) is among the first to visualize reward surfaces to characterize problems with policy gradient estimates. They plotted policy gradient estimates versus uniform random directions, demonstrating through striking visual examples that low-sample estimates of the policy gradient often guide the policy in directions no better than random ones. This work emphasizes the need to better understand the optimization landscape of rewards.

With the visualization methods by Li et al. (2018), Bekci & Gümüş (2020) utilized loss landscape visualizations to study Soft Actor-Critic (SAC) agents (Haarnoja et al., 2018) trained on inventory optimization tasks. They investigated the impact of policy stochasticity and action smoothing on the curvature of the loss landscapes in several MuJoCo environments, providing empirical evidence of how different factors influence the optimization landscape in RL.

Ota et al. (2024) utilized loss landscape visualization methods to compare shallow and deep networks in RL. It demonstrates that deeper models perform poorly due to increased complexity and curvature in loss landscapes. With the visual intuitions, they developed methods to effectively train deeper networks for RL tasks. Notably, their work plotted the loss function of SAC agents, which includes an entropy regularization term (Haarnoja et al., 2018). Recently, it is observed that an RL method with a flatter reward surface tends to show robust performance (Sullivan et al., 2022).

In our work, we also visualize the surface of rewards to confirm the flatness of reward functions. In addition, we compute the flatness metric to provide quantitative results.

## 3 PRELIMINARIES

This section outlines the key preliminaries that underpin our work, including the basics of Markov Decision Process (MDP), Policy Gradient Methods, Robust MDP, and Action Robust MDP. In addition, we briefly introduce Sharpness-Aware Minimization (SAM), which is a widely-documented optimization method to pursue flatter minima in loss surface for the supervised learning case.

### 3.1 MARKOV DECISION PROCESSES (MDP) AND POLICY GRADIENT METHODS

As well-documented in many works, MDP and policy gradient methods are formalized as follows. Our description mainly refers to (Sutton & Barto, 2018; Sutton et al., 1999; Williams, 1992).

**Markov Decision Process (MDP)** is defined by a tuple $(S, A, P, r, \gamma)$, where $S$ represents the set of states, $A$ represents the set of actions, $P : S \times A \times S \to [0, 1]$ is the state transition probability function, $r : S \times A \to \mathbb{R}$ is the reward function that assigns a real value to each state-action pair, and $\gamma \in [0, 1]$ is the discount factor that models the decreasing importance of future rewards.

The objective in an MDP is to find a policy $\pi : S \to A$ that maximizes the expected return. The expected return from a state $s$ under a policy $\pi$ is given by:

$$V^\pi(s) = \mathbb{E}\left[\sum_{t=0}^{\infty} \gamma^t r(s_t, a_t) \mid s_0 = s, \pi\right],$$

where $s_t$ and $a_t$ denote the state and action at time $t$, respectively, and the expectation is over the distribution of possible trajectories generated by following $\pi$.

**Policy Gradient Methods** are a crucial class of algorithms in deep reinforcement learning used to optimize policies directly. These methods employ gradients of the expected return with respect to the policy parameters to perform optimization (Sutton et al., 1999). In this case, the policy is commonly a parameterized model, i.e., $\pi_\theta(a|s)$, with learnable weights $\theta$. According to the *policy gradient theorem*, the gradient of the expected return with respect to the policy parameters $\theta$ is:

$$\nabla_\theta V^\pi(s) = \mathbb{E}_\pi\left[\sum_{t=0}^{\infty} \gamma^t \nabla_\theta \log \pi_\theta(a_t|s_t) R_t\right],$$

where $R_t = \sum_{k=t}^{\infty} \gamma^{k-t} r(s_k, a_k)$ denotes the cumulative reward from time $t$ onwards. This expression facilitates the application of gradient ascent methods to iteratively improve the policy by adjusting parameters in the direction that maximizes the expected return (Williams, 1992).

### 3.2 ROBUST MDPS AND ACTION ROBUST MDP

**Robust MDPs** extend the classical Markov Decision Process framework to handle uncertainties in transition probabilities and rewards (Nilim & El Ghaoui, 2005). In a robust MDP, instead of a single deterministic transition probability, the model allows for a range of possible outcomes defined by an uncertainty set $\mathcal{P}$. This set encapsulates all plausible variations in the transition dynamics under real-world conditions, and is defined as: $\mathcal{P} = \{\{p_{s,a}\}_{s \in S, a \in A} \mid p_{s,a} \in \mathcal{P}_{s,a}, \quad \forall s \in S, \forall a \in A\}$, where each $\mathcal{P}_{s,a}$ represents a compact subset of $\Delta_S$, the simplex of probability distributions over the state

space $S$, which is defined as: $\Delta_S = \left\{ p \in \mathbb{R}^{|S|} \mid p(s') \geq 0 \text{ for all } s' \in S, \sum_{s' \in S} p(s') = 1 \right\}$. The optimization problem in a robust MDP is framed as follows:

$$\max_{\pi} \min_{p \in \mathcal{P}} \mathbb{E}_{p,\pi} \left[ \sum_{t=0}^{\infty} \gamma^t r(s_t, a_t) \right], \tag{1}$$

focusing on a Max-Min strategy to ensure the policy is resilient against the worst-case scenario of transition probabilities within $\mathcal{P}$ (Iyengar, 2005).

**Action Robust MDP** specifically targets robustness in the context of actions taken within the MDP framework (Tessler et al., 2019). It considers the worst-case scenario where actions taken deviate from those chosen by the policy, potentially due to errors or external disturbances. Thus, the goal is to devise a policy that yields the highest minimum reward possible under any allowable perturbation of actions. Formally, the objective is as follows:

$$\max_{\pi} \min_{\|\delta_t\| \leq \beta} \mathbb{E}_{p,\pi} \left[ \sum_{t=0}^{\infty} \gamma^t r(s_t, a_t + \delta_t) \right], \tag{2}$$

where $\delta_t$ denotes allowable deviations in action space, upper bounded by $\beta \geq 0$, and $\gamma$ is the discount rate. This setup not only enhances the resilience of the strategy under practical conditions where perfect action execution is unfeasible but also shows the formulation with that used in Robust MDPs by maintaining the Max-Min structure across transition probability $p$ and reward function $r$.

### 3.3 SHARPNESS-AWARE MINIMIZATION (SAM)

**Sharpness-Aware Minimization (SAM)** aims to search loss minima that are not only high-performing but also demonstrating the minimal sensitivity to perturbations in the parameter space (Foret et al., 2021). The objective function of SAM is given by:

$$\min_{\theta} \max_{\|\epsilon\| \leq \rho} \mathcal{L}(\theta + \epsilon), \tag{3}$$

where $\mathcal{L}(\theta)$ is the loss function for a given batch of samples, $\theta$ represents the model parameters, and $\epsilon$ is the parameter perturbations within a sphere by the norm $\rho$. The Max operation inside the objective function seeks the worst-case performance within the perturbation boundary, and the outer Min operation attempts to find the model parameters that are most robust against such perturbations.

The gradient of the Min-Max problem can be approximated by four steps of computations. A detailed optimization procedure is provided in Appendix C.: i) Compute the gradient at $\theta$, which maximally increases the loss values, ii) Perturb model parameter $\theta$ to become $\theta + \epsilon$, where $\epsilon$ is pointing to the previously computed gradient direction. iii) Compute the gradient at the perturbed parameter $\theta + \epsilon$. iv) Update $\theta$ by using the gradient computed at $\theta + \epsilon$.

## 4 LINKING FLAT REWARD TO ACTION ROBUSTNESS

For a reward function $r(s, a)$ with a given state-action pair $(s, a)$, we here define the $\mathcal{E}$-flat reward maxima by rehearsing the definition of loss flatness of the supervised learning case (Shi et al., 2021).

**Definition 1** ($\mathcal{E}$-flat reward maxima) *For a reward function $r(s, a)$ and a policy model $\pi_\theta(a|s)$ parameterized by $\theta$, a maximum $\theta^*$ is $\mathcal{E}$-flat reward maxima when the following constraints hold:*

$$\text{For all } \epsilon \in \mathbb{R}^m \text{ s.t. } \|\epsilon\| \leq \mathcal{E}, \quad \mathbb{E}_{s \sim p, a \sim \pi_{\theta^* + \epsilon}(a|s)} \left[ \sum_{t=0}^{\infty} \gamma^t r(s_t, a_t) \right] = r^*$$

$$\text{There exists } \epsilon \in \mathbb{R}^m \text{ s.t. } \|\epsilon\| > \mathcal{E}, \quad \mathbb{E}_{s \sim p, a \sim \pi_{\theta^* + \epsilon}(a|s)} \left[ \sum_{t=0}^{\infty} \gamma^t r(s_t, a_t) \right] < r^* \tag{4}$$

*where $r^* := \mathbb{E}_{s \sim p, a \sim \pi_{\theta^*}(a|s)} \left[ \sum_{t=0}^{\infty} \gamma^t r(s_t, a_t) \right]$ and $\mathcal{E}$ is a positive real number.*

Based on the objective of the action robustness, we further define $\Delta$-action robust policy as follows:

**Definition 2** ($\Delta$-action robust policy) *For a reward function $r(s, a)$, a policy model $\pi_{\theta^*}(a|s)$ parameterized by $\theta^*$ is $\Delta$-action robust when the following constraints hold:*

$$\text{For all } \delta_t \in \mathbb{R}^{|A|} \text{ s.t. } \|\delta_t\| \leq \Delta, \quad \mathbb{E}_{s \sim p, a \sim \pi_{\theta^*}} \left[ \sum_{t=0}^{\infty} \gamma^t r(s_t, a_t + \delta_t) \right] = r^*$$

$$\text{There exists } \delta_t \in \mathbb{R}^{|A|} \text{ s.t. } \|\delta_t\| > \Delta, \quad \mathbb{E}_{s \sim p, a \sim \pi_{\theta^*}} \left[ \sum_{t=0}^{\infty} \gamma^t r(s_t, a_t + \delta_t) \right] < r^*, \tag{5}$$

*where $r^* := \mathbb{E}_{s \sim p, a \sim \pi_{\theta^*}} \left[ \sum_{t=0}^{\infty} \gamma^t r(s_t, a_t) \right]$ and $\Delta$ is a positive real number.*

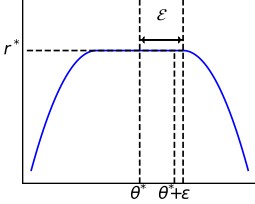
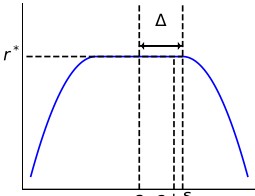

(a) $\mathcal{E}$-flat reward maxima          (b) $\Delta$-action robust policy

Figure 2: Illustration of Definitions 1 and 2. (a) $\mathcal{E}$-flat reward maxima as defined in Definition 1, highlighting the $\mathcal{E}$-flat reward $r^*$ region around $\theta^*$. (b) $\Delta$-action robust policy as defined in Definition 2, demonstrating how the policy maintains $r^*$ within perturbations of size $\Delta$.

Built upon the definitions, we here address how the flat reward maxima link to the action robust policy. To formalize the rigorous link, we provide a mathematical statement as follows.

**Proposition 1** *(Flat reward links to action robustness) If $\theta^*$ is an $\mathcal{E}$-flat reward maximum, then the policy $\pi_{\theta^*}$ is $\Delta^*$-action robust, where:*

$$\Delta^* \leq \|J(\theta^*)\|\mathcal{E} + \mathcal{O}(\mathcal{E}^2), \tag{6}$$

*and $J(\theta^*) := \nabla_\theta \mu_\theta(s)\big|_{\theta=\theta^*}$ is the Jacobian matrix of the mean action $\mu_\theta(s)$ with respect to $\theta$, evaluated at $\theta^*$.*

The proof of the proposition is in Appendix A.

**Remark 1.1** *(A link to Max-Min problem of action robustness) For $\Delta^*$-action robust policy derived by $\mathcal{E}$-flat reward maxima $\theta^*$, the policy directly satisfies the objective of action robust MDP:*

$$\theta^* = \arg\max_\theta \min_{\|\delta_t\|\leq\Delta^*} \mathbb{E}_{s\sim p, a\sim\pi_\theta}\left[\sum\nolimits_{t=0}^\infty \gamma^t r(s_t, a_t + \delta_t)\right], \tag{7}$$

*which implies that flatter reward yields the robustness against action perturbations.*

**Remark 1.2** *(An informal link to other factors) We here provide an intuition that the reward surface flatness also links to the robustness against the perturbations of other factors, including reward function and transition probability. i) For reward function perturbations, i.e., $r(s, a) \to \tilde{r}(s, a)$, it directly corresponds to the perturbations of loss function in the supervised learning case. Thus, when a reward function has merely slight perturbations, a maximum located at flatter surface tends to retain higher rewards. ii) When the MDP's transition probability has perturbations, i.e., $P(s'|s, a) \to \tilde{P}(s'|s, a)$, even the same action-state pair results in the different transition to other states. When the state transition changes, it gives different reward values, thus it relates to the undesired perturbation of rewards even with the same state-action pair. Consequently, it implies that reward flatness also promotes the robustness against the transition probability changes, which eventually links to the reward changes.*

## 5 EXPERIMENTAL RESULTS

**MuJoCo tasks:** We conduct extensive experiments across three continuous control tasks from the MuJoCo environment (Todorov et al., 2012): **HalfCheetah-v3**, **Hopper-v3**, and **Walker2d-v3**.

**Algorithms to be considered:** The evaluations are mainly for comparing the standard **PPO** algorithm and our SAM-enhanced PPO, which is called **SAM+PPO**. SAM+PPO's objective is:

$$\min_\theta \max_{\|\epsilon\|\leq\rho} \mathbb{E}_{p,\pi_{\theta+\epsilon}}\left[\sum\nolimits_{t=0}^\infty -\gamma^t r(s_t, a_t)\right], \tag{8}$$

where it solves the Min-Max problem with the perturbation $\epsilon$ that maximally worsens the reward, i.e., maximizing the minus-signed reward. Our SAM+PPO refers to the prior works (Lee

et al., 2024). Also, we add a recent baseline of robust reinforcement learning: **Robust Natural Actor-Critic (RNAC)** (Zhou et al., 2024) and **Robust Adversarial Reinforcement Learning (RARL)** (Pinto et al., 2017).

**Experiments to be done:** We evaluate PPO, RNAC, RARL, and SAM+PPO under various perturbations, including action noise, transition probability changes, and reward function variations. Also, we visualize the reward surfaces to analyze the flatness achieved by PPO and SAM+PPO and compare our method with existing robust reinforcement learning algorithms. Additional evaluations of other RL algorithms and SAM-enhanced version are presented in Appendix D.2

## 5.1 EXPERIMENTAL SETUP

For PPO, SAM+PPO, RNAC, and RARL, we use a multi-layer perceptron (MLP) architecture with three layers for their policy networks. Also, for each experimental case, PPO and SAM+PPO are trained in the nominal (unperturbed) environment using identical hyperparameters for a fair comparison. For RNAC and RARL, we tune the hyperparameters to achieve the performance reported in (Zhou et al., 2024), and (Pinto et al., 2017) respectively, specified in the original RNAC and RARL implementation for the environments. Each experiment was conducted over five independent trials, each initialized with a different random seed to ensure statistical significance. Furthermore, for each evaluation, we performed 100 evaluation runs and averaged the results to enhance the stability and accuracy of our findings. We added the detailed hyperparameters and settings in Appendix B.

## 5.2 ACTION ROBUSTNESS EVALUATION

We first evaluate the action robustness, which seems to be the most highly related factor to flat rewards. To simulate the perturbed actions, we add zero-mean Gaussian noise with varying standard deviations $\sigma_a$ ranging from 0.0 to 0.5 to the agent's actions. The noisy action $a_{\text{noisy}}$ is computed as: $a_{\text{noisy}} = a + \mathcal{N}(0, \sigma_a^2)$, where $a$ is the original action output by the agent's policy. If the addition of noise results in actions outside the valid range, we clip the actions to the allowable bounds of the environment.

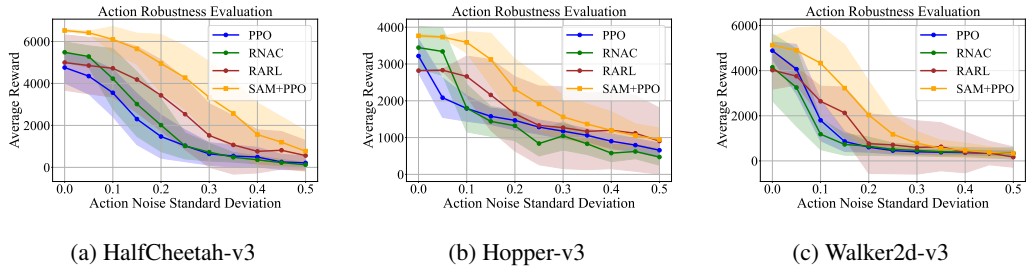

| (a) HalfCheetah-v3 | (b) Hopper-v3 | (c) Walker2d-v3 |

Figure 3: Action robustness evaluation across different environments. The average return is plotted against the action noise standard deviation $\sigma_a$.

Figure 3 presents the results of the action robustness evaluation across three distinct environments: HalfCheetah-v3, Hopper-v3, and Walker2d-v3. In all three environments, the performance of all agents gradually decreases as the action noise increases, which is expected due to the increased uncertainty in action execution. Importantly, SAM+PPO consistently outperforms PPO and RNAC across all noise levels, and the rate of performance degradation is much slower compared to PPO and RNAC. RARL demonstrates competitive robustness performance, maintaining relatively higher returns compared to PPO and RNAC under moderate noise levels. These results indicate that the flat reward achieved by SAM+PPO makes the policy less sensitive to action perturbations, thus coinciding with our claim.

## 5.3 TRANSITION PROBABILITY ROBUSTNESS EVALUATION

We perturb the transition dynamics by modifying the physical parameters of the environment. By tailoring to the given three environments, we vary the *torso mass* and *friction coefficients*.

**Variation in Torso Mass:** We scale the torso mass of the agent by factors ranging from 0.5 to 1.5 times its nominal value in increments of 0.1. This tests the agent's ability to adapt to changes in its own dynamics, such as carrying additional weight or structural modifications.

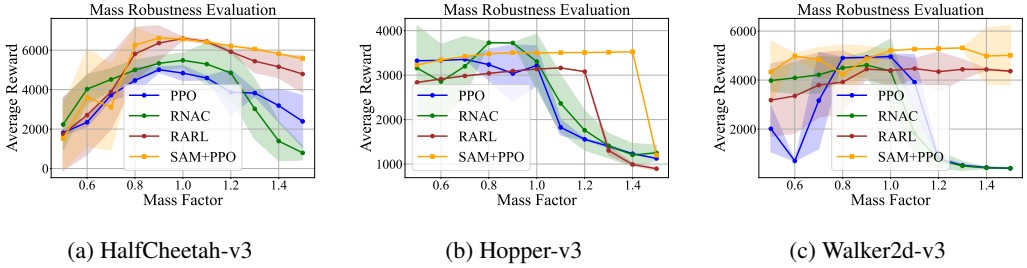

|     |     |     |
| :-: | :-: | :-: |
| (a) HalfCheetah-v3 | (b) Hopper-v3 | (c) Walker2d-v3 |

Figure 4: Robustness evaluation under torso mass variations. The average return is plotted against the mass scaling factor. The nominal torso mass factor is 1.0.

As shown in Figure 4, the performance of all agents decreases as the torso mass deviates from the nominal. However, SAM+PPO consistently achieves higher returns than others. RNAC generally shows better performance than PPO but worse than SAM+PPO. RARL demonstrates competitive performance, maintaining relatively higher returns compared to PPO and RNAC under moderate mass scaling factors. These results suggest that the policies learned by SAM+PPO are more robust to changes in the agent's dynamics, likely due to the flatter reward surface that reduces sensitivity to such perturbations.

**Variation in Friction Coefficient:** We vary the friction coefficients between the agent's feet and the ground from 0.4 to 1.6. The nominal friction coefficient is 1.0. This simulates different ground conditions, such as slippery or rough surfaces, which affect the agent's ability to move and balance.

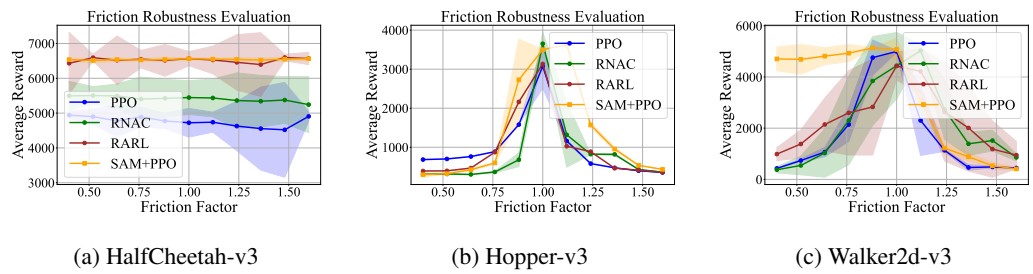

|     |     |     |
| :-: | :-: | :-: |
| (a) HalfCheetah-v3 | (b) Hopper-v3 | (c) Walker2d-v3 |

Figure 5: Robustness evaluation under friction coefficient variations. The average return is plotted against the friction scaling factor.

Figure 5 presents the friction robustness evaluation. SAM+PPO shows a clear advantage over PPO, maintaining higher performance as the friction coefficient varies. SAM+PPO generally outperforms RNAC. RNAC shows strong performance with larger friction factors for Walker2d-v3, but it severely degrades when the friction decreases (less friction makes the ground surface slippery). RARL maintains relatively stable performance across a range of friction coefficients, particularly excelling in environments with extreme cases of friction where other methods experience significant degradation. These results indicate that the policies learned by SAM+PPO are more adaptable to different surface conditions, which is critical for real-world applications where ground properties can change unpredictably.

**Mass and Friction Joint Variations:** Herein, we focus on evaluating the agents' performance across the joint variations of mass and friction, which shows a comprehensive view of robustness.

Figure 6 presents the reward heatmaps for three environments. The widely highlighted regions of SAM+PPO demonstrate that it is robust compared to others across a broader spectrum of environmental variations. While SAM+PPO shows slightly lower performance in scenarios with extremely high friction, it significantly outperforms others in environments with lower friction, such as slippery surfaces. RARL exhibits competitive performance in certain regions, particularly maintaining

better performance than PPO and RNAC under specific combined perturbations of mass and friction. Surprisingly, RNAC shows marginal gains over PPO. It underscores the superior robustness of SAM+PPO in adapting to compounded changes in mass and friction over others.

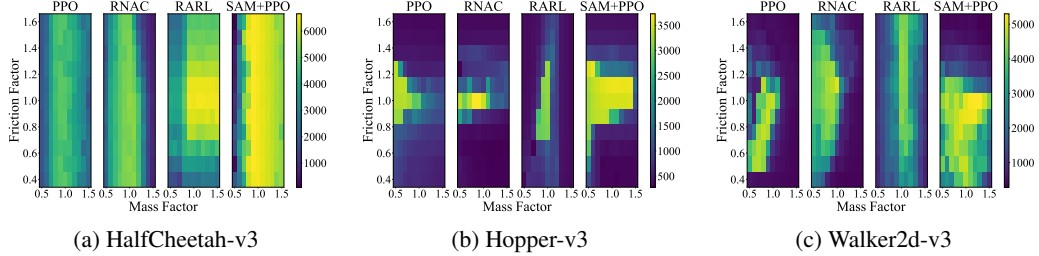

| (a) HalfCheetah-v3 | (b) Hopper-v3 | (c) Walker2d-v3 |

Figure 6: Reward heatmaps for combined mass and friction variations across different environments. Yellow to dark blue indicates better to worse performance, respectively.

Table 1 summarizes the performance of PPO, SAM+PPO, RNAC and RARL under various perturbation scenarios across different environments. These results collectively demonstrate that SAM+PPO enhances the agent's ability to withstand simultaneous environmental perturbations in mass and friction, providing a more robust and reliable performance compared to others.

Table 1: Performance comparison under various perturbations.

| Perturbation | Metric | HalfCheetah-v3 | | Hopper-v3 | | Walker2d-v3 | |
|---|---|---|---|---|---|---|---|
| | | Nominal | Perturbed | Nominal | Perturbed | Nominal | Perturbed |
| **Action Noise** $\sigma = 0.2$ | PPO | 4758 | 1469(−3289) | 3217 | 1467(−1750) | 4883 | 607(−4276) |
| | RNAC | 5484 | 2014(−3470) | 3445 | 1321(−2124) | 4147 | 652(−3495) |
| | RARL | 4996 | 3412(−1584) | 2819 | **1645(−1174)** | 4020 | 764(−3256) |
| | SAM+PPO | **6523** | **4949(−1574)** | **3766** | 2312(−1454) | **5129** | **2033(−3096)** |
| **Mass Scale Factor** 1.2 | PPO | 4837 | 3865(−972) | 3215 | 1556(−1659) | 4957 | 782(−4175) |
| | RNAC | 5485 | 4844(−641) | 3303 | 1759(−1544) | 4373 | 741(−3632) |
| | RARL | 6561 | 6016(−545) | 3136 | 3078(−58) | 4393 | 4343(−50) |
| | SAM+PPO | **6562** | **6210(−352)** | **3499** | **3508(+9)** | **5205** | **5284(+79)** |
| **Friction Coefficient** 0.88 | PPO | 4723 | **4774(+51)** | 3075 | 1580(−1495) | 4996 | 4756(−240) |
| | RNAC | 5446 | 5422(−24) | **3653** | 679(−2974) | 4438 | 3844(−594) |
| | RARL | 6560 | 6516(−44) | 3136 | 2161(−975) | 4426 | 2828(−1598) |
| | SAM+PPO | **6562** | 6539(−23) | 3500 | **2728(−772)** | **5073** | **5134(+61)** |

(−/+) values mean the performance degradation from 'Nominal' to 'Perturbed.',
Boldface is used for highest performance in 'Nominal', smallest performance degradation in 'Perturbed'

## 5.4 REWARD FUNCTION ROBUSTNESS EVALUATION

For testing reward function robustness, a notable difference of this part is that reward perturbations are added during training. As done in the previous testing, when the training is done with nominal rewards and then perturbed during testing, the policy thus yields the expectation of the perturbed rewards, leading to a trivial message. Therefore, we adopt noise rewards during training to confirm how different agents are trained against the reward perturbations. Specifically, we introduced zero-mean Gaussian noise with a standard deviation of $\sigma_r = 0.1$ to the rewards. After training, we evaluated the policies in the nominal environment.

Table 2: Performance comparison of agents trained with and without reward noise ($\sigma_r = 0.1$)

| Algorithm | HalfCheetah-v3 | | Hopper-v3 | | Walker2d-v3 | |
|---|---|---|---|---|---|---|
| | Nominal | Noisy | Nominal | Noisy | Nominal | Noisy |
| PPO | 4820 | 3688(−1132) | 3150 | 2945(−205) | 4780 | 2204(−2576) |
| RNAC | 5423 | 4088(−1335) | 3211 | 3035(−176) | 4184 | 3172(−1012) |
| RARL | 5620 | 4617(−1003) | 3124 | 2993(−131) | 4388 | 3085(−1303) |
| SAM+PPO | **6530** | **5990(−540)** | **3505** | **3377(−128)** | **5120** | **4226(−894)** |

(−) values means the performance degradation from 'Nominal' to 'Noisy.'

As shown in Table 2, the degradation of SAM+PPO is significantly less than that of PPO across all cases. SAM+PPO is less sensitive to variations of rewards, leading to better stability. It is critical when the reward may not always be accurate or consistent in real-world settings.

## 5.5 REWARD SURFACE VISUALIZATIONS AND FLATNESS MEASUREMENTS

We visualize the reward surface by using the method in (Sullivan et al., 2022). Also, we first compute the flatness metrics, including the maximum eigenvalue of the Hessian, i.e., $\lambda_{\max}$ and the Low-Pass Filter (LPF) flatness measure (Keskar et al., 2017; Bisla et al., 2022).

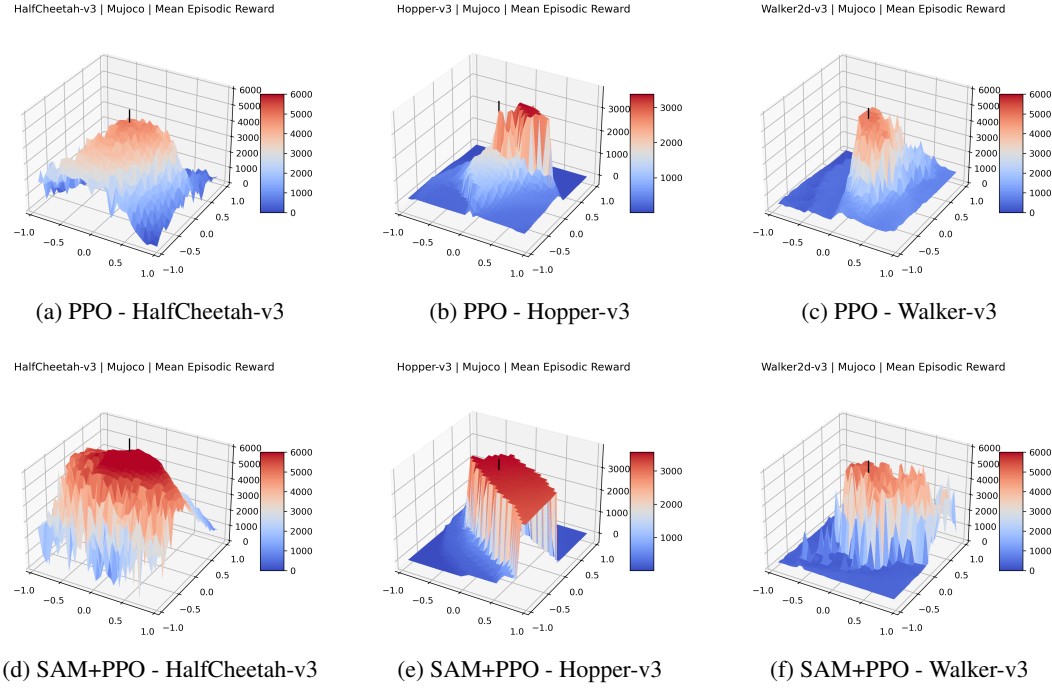

(a) PPO - HalfCheetah-v3     (b) PPO - Hopper-v3     (c) PPO - Walker-v3

(d) SAM+PPO - HalfCheetah-v3     (e) SAM+PPO - Hopper-v3     (f) SAM+PPO - Walker-v3

Figure 7: Reward surface visualizations of agents. The x and y axes represent perturbations along random directions in the parameter space, and the z-axis represents the average return.

Figure 7 shows the reward surfaces for PPO and SAM+PPO in the HalfCheetah-v3 environment. Similar patterns are observed in Hopper-v3 and Walker2d-v3. The SAM+PPO reward surface is noticeably flatter, meaning that the agent's performance is less sensitive to parameter changes. This supports that SAM yields flatter reward maxima, and it is strongly correlated to more robust policies.

As shown in Table 3, SAM+PPO generally achieves lower flatness metrics compared to PPO, indicating that SAM+PPO converges to flatter minima in the policy parameter space.

Table 3: Flatness metrics for PPO and SAM+PPO (↓: indicates that lower is better).

| Metrics | $\lambda_{\max}$ ↓ (Keskar et al., 2017) | | | LPF ↓ (Bisla et al., 2022) | | |
|---|---|---|---|---|---|---|
| Environment | HalfCheetah-v3 | Hopper-v3 | Walker2d-v3 | HalfCheetah-v3 | Hopper-v3 | Walker2d-v3 |
| PPO | 15192.95 | 131.07 | 7239.59 | 0.0385 | 0.00034 | 0.0269 |
| SAM+PPO | 275.93 | 80.86 | 271.91 | 0.00097 | 0.00018 | 0.00028 |

## 6 CONCLUSION

In this paper, we aim to unravel the meaning of flat reward maxima from the perspective of the robustness of reinforcement learning. Theoretically, we provide a formal link between flatness and robustness. Empirically, we broadly confirm our claim, showing that SAM+PPO outperforms standard PPO and existing robust RL algorithms like RARL and RNAC, validating the efficacy of flat rewards. This work emphasizes the importance of flatness in RL's policy space. We believe that it can broaden new avenues for developing robust RL algorithms by combining with prior robust RL methodologies to pursue a strongly robust RL in real-world applications.

ACKNOWLEDGMENTS

This work was supported by the Institute of Information & communications Technology Planning & Evaluation (IITP) grant funded by the Korea government (MSIT) (No. RS-2020-II201336, Artificial Intelligence Graduate School Program (UNIST)), (No. IITP-2025-RS-2022-00156361, Innovative Human Resource Development for Local Intellectualization program), and the National Research Foundation of Korea (NRF) grant funded by the Korea government (MSIT) (No. RS-2024-00459023).

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

# A  PROOF OF PROPOSITION 1

We aim to show that an $\mathcal{E}$-flat reward maximum $\theta^*$ (as per Definition 1) corresponds to a $\Delta^*$-action robust policy $\pi_{\theta^*}$ (as per Definition 2), with:

$$\Delta^* \leq \|J(\theta^*)\|\mathcal{E} + \mathcal{O}(\mathcal{E}^2), \tag{9}$$

where $J(\theta^*) = \nabla_\theta \mu_\theta(s)\big|_{\theta=\theta^*}$ is the Jacobian matrix of the mean action $\mu_\theta(s)$ with respect to $\theta$, evaluated at $\theta^*$.

Consider a stochastic policy $\pi_\theta(a|s)$ parameterized by $\theta$, assumed to be a Gaussian policy with mean action $\mu_\theta(s)$ and fixed covariance matrix $\Sigma$:

$$\pi_\theta(a|s) = \mathcal{N}(a; \mu_\theta(s), \Sigma). \tag{10}$$

Let $\epsilon \in \mathbb{R}^m$ be a perturbation such that $\|\epsilon\| \leq \mathcal{E}$. Performing a first-order Taylor expansion around $\theta^*$:

$$\mu_{\theta^*+\epsilon}(s) = \mu_{\theta^*}(s) + J(\theta^*)\epsilon + \mathcal{O}(\|\epsilon\|^2). \tag{11}$$

Define the action perturbation:

$$\delta_t = \mu_{\theta^*+\epsilon}(s_t) - \mu_{\theta^*}(s_t) = J(\theta^*)\epsilon + \mathcal{O}(\|\epsilon\|^2). \tag{12}$$

Then,

$$\|\delta_t\| \leq \|J(\theta^*)\|\|\epsilon\| + \mathcal{O}(\|\epsilon\|^2) \leq \|J(\theta^*)\|\mathcal{E} + \mathcal{O}(\mathcal{E}^2). \tag{13}$$

From Definition 1, for $\|\epsilon\| \leq \mathcal{E}$:

$$\mathbb{E}_{s\sim p, a\sim \pi_{\theta^*+\epsilon}}\left[\sum_{t=0}^{\infty} \gamma^t r(s_t, a_t)\right] = r^*. \tag{14}$$

Noting that shifting the mean of the Gaussian policy is equivalent to shifting the action variable, we have:

$$\pi_{\theta^*+\epsilon}(a_t|s_t) = \pi_{\theta^*}(a_t - \delta_t|s_t). \tag{15}$$

Thus, the expected cumulative reward under the perturbed policy becomes:

$$\mathbb{E}_{s\sim p, a\sim \pi_{\theta^*+\epsilon}}\left[\sum_{t=0}^{\infty} \gamma^t r(s_t, a_t)\right] = \mathbb{E}_{s\sim p, a_t\sim \pi_{\theta^*+\epsilon}}\left[\sum_{t=0}^{\infty} \gamma^t r(s_t, a_t)\right] \tag{16}$$

$$= \mathbb{E}_{s\sim p, a_t\sim \pi_{\theta^*}}\left[\sum_{t=0}^{\infty} \gamma^t r(s_t, a_t + \delta_t)\right]. \tag{17}$$

Since $\mathbb{E}_{s \sim p, a \sim \pi_{\theta^* + \epsilon}} \left[ \sum_{t=0}^{\infty} \gamma^t r(s_t, a_t) \right] = r^*$, it follows that:

$$\mathbb{E}_{s \sim p, a \sim \pi_{\theta^*}} \left[ \sum_{t=0}^{\infty} \gamma^t r(s_t, a_t + \delta_t) \right] = r^*. \tag{18}$$

This implies that for all $\|\delta_t\| \leq \Delta^*$ with $\Delta^* = \|J(\theta^*)\|\mathcal{E} + \mathcal{O}(\mathcal{E}^2)$, the expected cumulative reward remains $r^*$, satisfying the first condition of a $\Delta^*$-action robust policy as per Definition 2.

Moreover, since there exists $\epsilon$ with $\|\epsilon\| > \mathcal{E}$ such that $\mathbb{E}_{s,a} \left[ \sum_{t=0}^{\infty} \gamma^t r(s_t, a_t) \right] < r^*$, the corresponding $\delta_t$ satisfies $\|\delta_t\| > \Delta^*$, and thus the expected cumulative reward decreases below $r^*$ for perturbations larger than $\Delta^*$, fulfilling the second condition.

## A.1 Discussion on the Bounds of the Jacobian

In the proof of Proposition 1, we derive a bound on the action robustness parameter $\Delta^*$ in terms of the Jacobian $\|J(\theta^*)\|$ and the flatness parameter $\mathcal{E}$:

$$\Delta^* \leq \|J(\theta^*)\|\mathcal{E} + \mathcal{O}(\mathcal{E}^2).$$

We acknowledge that in practical applications, especially when using deep neural networks for policy representations, it is challenging to guarantee that the Jacobian $\|J(\theta^*)\|$ is bounded. Deep neural networks can have complex architectures and nonlinear activation functions that may lead to large gradients and, consequently, large Jacobian norms.

However, several common practices during neural network training help control the magnitude of the Jacobian:

- **Weight Regularization**: Techniques such as $L_2$ regularization (weight decay) penalize large weights, which indirectly constrains the Jacobian. By limiting the magnitude of the weights, the sensitivity of the network outputs to changes in the inputs and parameters is reduced.

- **Gradient Clipping**: Applying gradient clipping during optimization prevents the gradients from becoming excessively large. This helps maintain the stability of parameter updates and controls the growth of the Jacobian norm.

- **Bounded Activation Functions**: Using activation functions with bounded derivatives (e.g., hyperbolic tangent, sigmoid) limits the rate of change of the network outputs with respect to the inputs and parameters, thus contributing to bounding the Jacobian.

- **Normalization Techniques**: Methods like batch normalization and layer normalization can help stabilize the learning process and control the scale of activations and gradients, affecting the Jacobian norm.

Regarding the SAM objective, SAM promotes convergence to flatter regions in the loss landscape by considering adversarial perturbations of the network parameters during optimization with the objective function in Equation 3,

where $L(\theta)$ is the loss function, and $\rho$ defines the size of the neighborhood around the current parameters $\theta$. By optimizing for the worst-case loss within a neighborhood, SAM discourages sharp minima with high curvature, which are associated with large Hessian norms. This process inherently suppresses large gradients and promotes smoother variations of the loss with respect to parameter changes. While SAM primarily targets the curvature of the loss landscape (second-order information), this effect also influences the Jacobian of the network outputs with respect to the parameters (first-order information).

Our empirical results demonstrate that SAM+PPO leads to policies that are more robust to perturbations in both the parameters and the environment dynamics. This supports the practical relevance of our theoretical findings, suggesting that SAM effectively contributes to controlling the Jacobian norm in practice.

While it may not be possible to guarantee a bounded Jacobian everywhere in the network, the combination of SAM and common training practices helps in maintaining the Jacobian within a reasonable

range. This enhances the robustness of the learned policies and aligns with the conditions assumed in our theoretical analysis.

## B EXPERIMENTAL DETAILS

In this appendix, we provide detailed information about the hyperparameters and experimental settings used in our experiments.

### B.1 NETWORK ARCHITECTURE

For all agents, including PPO, SAM+PPO, and RNAC, we employ a multi-layer perceptron (MLP) architecture for both the actor (policy network) and the critic (value network). The network consists of an input layer matching the state dimension of the environment, followed by three fully connected hidden layers, each with 64 neurons and `Tanh` activation functions. The output layer of the actor network produces the parameters of the action distribution (mean and log standard deviation for Gaussian policies), while the critic network outputs a single scalar value representing the state value estimate. We apply orthogonal initialization to all layers to enhance training stability.

### B.2 HYPERPARAMETERS AND TRAINING SETTINGS

We use identical hyperparameters for PPO, SAM+PPO, and RNAC to ensure a fair comparison. The shared hyperparameters are as follows: the discount factor $\gamma$ is set to $0.99$, the GAE parameter $\lambda$ is $0.95$, and the PPO clip parameter $\epsilon$ is $0.2$. Both the actor and critic learning rates are set to $3 \times 10^{-4}$, with the Adam optimizer used for optimization. The batch size is $2048$, and the mini-batch size is $64$, with $10$ PPO epochs per update ($K_{\text{epochs}} = 10$). We employ gradient clipping with a maximum norm of $0.5$, learning rate decay, and state normalization for all agents.

For **SAM+PPO**, we introduce the SAM optimizer with $\rho$ parameters of $0.008$ for HalfCheetah-v3 and Walker2d-v3, and $0.01$ for Hopper-v3, which control the neighborhood size for sharpness-aware minimization. For **RNAC**, we adjust the uncertainty set to "IPM" to model uncertainties in the transition dynamics and configure the number of next steps to $2$, allowing the agent to consider multiple potential future states.

For comparison with existing robust reinforcement learning algorithms, we trained Robust Adversarial Reinforcement Learning (RARL) (Pinto et al., 2017) using the following hyperparameters specific to the Hopper-v3 environment. The batch size was set to $4000$ to align with the original RARL implementation, and the mini-batch size remained at $64$. The network's hidden width was increased to $100$ neurons per layer to match the architecture used in RARL. Both the protagonist and adversary actors and critics were trained with learning rates of $0.0003$. The discount factor $\gamma$ was set to $0.995$, and the GAE parameter $\lambda$ was set to $0.97$, consistent with the original RARL settings. The entropy coefficient was maintained at $0.0$, and weight regularization was applied with a parameter of $0.00005$. We set the adversarial fraction to $0.25$ to control the adversary's action strength and configured the number of iterations for both the protagonist and adversary to $1$. All other settings, including gradient clipping, learning rate decay, state normalization, and orthogonal initialization, were kept consistent with PPO, SAM+PPO, and RNAC to ensure a fair comparison.

## C DETAILS AND ANALYSIS OF SAM INTEGRATED WITH PPO

In this appendix, we provide a detailed discussion on the integration of SAM with RL, especially with PPO. We elaborate on the optimization process of the min-max objective, discuss the computational overhead introduced by SAM, and analyze the sensitivity to the perturbation radius $\rho$.

### C.1 OPTIMIZATION OF THE MIN-MAX OBJECTIVE

The SAM optimization modifies the standard PPO objective by introducing a maximization over parameter perturbations within an $\ell_2$ norm ball of radius $\rho$. For convenience, we restate the objective as presented in Equation 3 from Section 3.3:

$$\min_{\theta} \max_{\|\epsilon\|_2 \leq \rho} L(\theta + \epsilon), \tag{19}$$

where $L(\theta)$ is the PPO loss function, and $\theta$ represents the policy parameters.

### C.1.1 LINKAGE TO ACTION ROBUST MDP

In PPO, the loss function $L(\theta)$ is inherently tied to the optimization of the expected cumulative reward. Specifically, PPO aims to maximize the expected cumulative discounted reward by optimizing the policy parameters $\theta$. The PPO loss function can be expressed in relation to the reward function $r(s, a)$ as follows:

$$L(\theta) = -\mathbb{E}_{s \sim p, a \sim \pi_\theta} \left[ \sum_{t=0}^{\infty} \gamma^t r(s_t, a_t) \right], \tag{20}$$

where the negative sign indicates that minimizing $L(\theta)$ is equivalent to maximizing the expected cumulative reward. This formulation aligns with the reward-based learning paradigm of RL, where the objective is to find a policy that yields the highest expected cumulative reward across all possible state-action pairs.

With Equation 20, Equation 19 can be expressed as:

$$\min_{\theta} \max_{\|\epsilon\| \leq \rho} -\mathbb{E}_{s \sim p, a \sim \pi_{\theta+\epsilon}} \left[ \sum_{t=0}^{\infty} \gamma^t r(s_t, a_t) \right], \tag{21}$$

which aligns with Equation 8 in Section 5, incorporating the cumulative discounted reward into the expectation. Both formulations aim to optimize the policy parameters $\theta$ by considering worst-case perturbations within a specified norm bound.

Based on Proposition 1, Remark 1.1 of Section 4 and the corresponding proof in Appendix A, we derive that the SAM+PPO objective function aims to find the $\mathcal{E}$-flat reward maxima as defined in Definition 1. Consequently, this leads to the formulation of a $\Delta$-action robust policy as defined in Definition 2. Therefore, the objective function aligns with Equation 2 of the main text:

$$\max_{\pi} \min_{\|\delta_t\| \leq \beta} \mathbb{E}_{s \sim p, a \sim \pi} \left[ \sum_{t=0}^{\infty} \gamma^t r(s_t, a_t + \delta_t) \right], \tag{22}$$

where $\delta_t$ represents perturbations to the actions at time $t$, and $\beta$ is a scaling factor analogous to $\rho$. This formulation corresponds to an Action Robust MDP, where the policy seeks to maximize the expected cumulative reward while minimizing the impact of worst-case perturbations to actions within a specified norm bound.

### C.1.2 OPTIMIZATION PROCEDURE

Optimizing the min-max objective in Equation 19 involves integrating the SAM approach with the PPO framework. Below, we present the integrated algorithm and provide a detailed explanation of each step.

---

**Algorithm 1** SAM Integrated with PPO

---

**Require:** Initial policy parameters $\theta_0$, perturbation radius $\rho$, learning rate $\alpha$, clipping parameter $\epsilon$

1: **for** each iteration **do**
2:     **Collect Trajectories:** Use current policy $\pi_\theta$ to collect trajectories by interacting with the environment.
3:     **Compute Advantages:** Calculate advantages $\hat{A}_t$ and returns based on collected trajectories.
4:     **Compute PPO Loss:**

$$L(\theta) = \mathbb{E}\left[\min\left(r_t(\theta)\hat{A}_t, \text{clip}(r_t(\theta), 1 - \epsilon, 1 + \epsilon)\hat{A}_t\right)\right]$$

5:     **Compute Base Gradient:** $g_\theta = \nabla_\theta L(\theta)$
6:     **Apply SAM Perturbation:**

$$\epsilon^* = \rho\frac{g_\theta}{\|g_\theta\|_2}$$

7:     **Compute Perturbed Loss:** $L_{\text{perturbed}} = L(\theta + \epsilon^*)$
8:     **Compute Perturbed Gradient:** $g_{\theta+\epsilon^*} = \nabla_\theta L(\theta + \epsilon^*)$
9:     **Update Policy Parameters:** $\theta \leftarrow \theta - \alpha g_{\theta+\epsilon^*}$
10: **end for**=0

---

1. **Collect Trajectories:** $\pi_\theta \rightarrow$ Collect trajectories using $\pi_\theta$

   Using the current policy parameters $\theta$, we collect a batch of trajectories by interacting with the environment. These trajectories are essential for estimating the PPO loss.

2. **Compute Advantages:** $\hat{A}_t =$ Calculate advantages based on collected trajectories

   Advantages $\hat{A}_t$ are computed to reduce variance in the policy gradient estimates, enhancing the stability of training.

3. **Compute PPO Loss:** $L(\theta) = \mathbb{E}\left[\min\left(r_t(\theta)\hat{A}_t, \text{clip}(r_t(\theta), 1 - \epsilon, 1 + \epsilon)\hat{A}_t\right)\right]$

   The PPO loss function $L(\theta)$ incorporates a clipping mechanism to prevent large updates, ensuring stable policy optimization.

4. **Compute Base Gradient:** $g_\theta = \nabla_\theta L(\theta)$

   We compute the gradient of the PPO loss with respect to the policy parameters $\theta$. This gradient indicates the direction in which the loss increases most rapidly.

5. **Apply SAM Perturbation:** $\epsilon^* = \rho\frac{g_\theta}{\|g_\theta\|_2}$

   While the inner maximization problem theoretically considers all possible directions within the $\rho$-ball around $\theta$, choosing $\epsilon$ in the direction of the gradient $g_\theta$ is a first-order approximation that efficiently captures the worst-case perturbation. This approximation is rooted in the Taylor expansion of the loss function, where the gradient direction signifies the direction of maximum loss increase. By perturbing $\theta$ in this direction, we effectively approximate the maximum increase in loss within the allowed perturbation magnitude $\rho$.

6. **Compute Perturbed Loss:** $L_{\text{perturbed}} = L(\theta + \epsilon^*)$

   An additional forward pass is conducted using the perturbed parameters $\theta + \epsilon^*$ to evaluate the loss under perturbed conditions. This step assesses how the loss behaves when the parameters are subjected to the worst-case perturbation identified in the previous step.

7. **Compute Perturbed Gradient:** $g_{\theta+\epsilon^*} = \nabla_\theta L(\theta + \epsilon^*)$

   We compute the gradient of the perturbed loss with respect to the policy parameters. This gradient informs the parameter update step by indicating how to adjust $\theta$ to minimize the worst-case loss.

8. **Update Policy Parameters:** $\theta \leftarrow \theta - \alpha g_{\theta+\epsilon^*}$

   Where $\alpha$ is the learning rate. The policy parameters $\theta$ are updated using the gradient of the perturbed loss, thereby incorporating the robustness introduced by SAM. This update step moves the parameters in the direction that minimizes the worst-case loss, enhancing the policy's robustness against adversarial perturbations.

This optimization procedure effectively seeks parameters that minimize the worst-case loss within a neighborhood of radius $\rho$ around the current parameters $\theta$, thereby enhancing the robustness of the policy.

## C.2 COMPUTATIONAL OVERHEAD ANALYSIS

Integrating SAM with PPO introduces additional computational overhead due to the extra forward and backward passes required to compute the perturbation and update the policy parameters using the perturbed loss.

### C.2.1 THEORETICAL ANALYSIS

In standard PPO, each optimization step involves one forward pass and one backward pass to compute the gradient $\nabla_\theta L(\theta)$. With SAM, the optimization step involves:

- **First Forward and Backward Pass:** Compute $g_\theta = \nabla_\theta L(\theta)$.
- **Second Forward Pass:** Compute $L(\theta + \epsilon^*)$ with perturbed parameters.
- **Second Backward Pass:** Compute $g_{\theta+\epsilon^*} = \nabla_\theta L(\theta + \epsilon^*)$.

This effectively doubles the number of forward and backward passes per optimization step. In Big $\mathcal{O}$ notation, the per-iteration computational complexity increases from $\mathcal{O}(N)$ for PPO to $\mathcal{O}(2N)$ for SAM+PPO, where $N$ is the number of parameters.

### C.2.2 EMPIRICAL ANALYSIS

We empirically measured the training time per iteration for both PPO and SAM+PPO. The results are summarized in Table 4.

Table 4: Comparison of agents Training Time (sec) per Iteration across Environments

| Algorithm | HalfCheetah-v3 | Hopper-v3 | Walker2d-v3 |
|---|---|---|---|
| PPO | 1.22 | 0.13 | 0.2 |
| SAM+PPO | 1.5($\times$1.83) | 0.23($\times$1.76) | 0.24($\times$1.20) |

The results show that SAM+PPO incurs approximately a 61% increase in training time per iteration compared to PPO. Considering the significant gains in robustness, this overhead is a reasonable trade-off. While the per-iteration training and update times for SAM+PPO are higher compared to standard PPO, the overall training time presents a more favorable comparison. Specifically, as shown in Table 9, SAM+PPO achieves comparable or slightly increased overall training times while converging faster and attaining rewards more efficiently. This indicates that SAM+PPO not only enhances robustness and generalization but also maintains efficient training dynamics.

## C.3 SENSITIVITY TO HYPERPARAMETER $\rho$

The perturbation radius $\rho$ introduced by SAM is a critical hyperparameter that controls the extent of parameter perturbations during optimization. Selecting an appropriate value for $\rho$ is crucial for balancing robustness and training stability.

- **Large $\rho$:** Encourages the optimizer to find flatter maxima, potentially enhancing robustness. However, if $\rho$ is too large, it may lead to training instability or convergence issues.

- **Small $\rho$:** May result in less robustness gain, as the perturbations are insufficient to promote significant flatness in the parameter space.

We conducted experiments by varying $\rho$ within a practical range to assess its impact on performance. The values tested were $\rho \in \{0.001, 0.005, 0.01, 0.05, 0.1\}$. Figure 8 illustrates the performance of SAM+PPO with different values of $\rho$ on the Hopper-v3 environment.

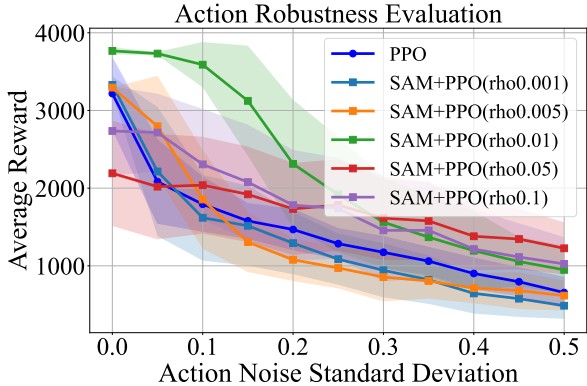

Figure 8: Sensitivity of SAM+PPO performance to the perturbation radius $\rho$ in Hopper-v3 environment.

- **Optimal Range:** A value of $\rho = 0.01$ provided the best results of robustness and training stability.

- **Small $\rho$:** At $\rho = 0.001, 0.005$, the robustness gains were minimal compared to standard PPO.

- **Large $\rho$:** At $\rho = 0.05, 0.1$, training became unstable, and performance degraded significantly.

### C.3.1 EFFECT ON PPO HYPERPARAMETERS

The integration of SAM into PPO modifies the optimization process by adding an outer loop that seeks parameters robust to perturbations within a specified radius $\rho$. This outer loop operates independently of PPO's internal mechanisms. Since SAM's perturbation process focuses on finding flatter regions in the parameter space without altering PPO's update equations or loss function structure, it does not inherently change the role or sensitivity of PPO's original hyperparameters.

The perturbation radius $\rho$ is a pivotal hyperparameter in SAM+PPO that directly influences the balance between robustness and training stability. Through our sensitivity analysis, we identified $\rho = 0.05$ as the optimal value that offers significant robustness gains without compromising training stability. Additionally, integrating SAM does not necessitate retuning of PPO's existing hyperparameters, making the combination both effective and practical.

## D  ADDITIONAL EXPERIMENTAL RESULTS

### D.1  EXPERIMENTS ON DISCRETE ACTION ENVIRONMENTS

In the main text, our experiments focused on continuous control environments. To validate the applicability and reliability of our SAM-enhanced method in a broader context, we extended our experiments to discrete action environments provided by OpenAI Gym: CartPole-v1 and LunarLander-v2. These environments allow us to assess the performance of SAM+PPO in settings with discrete action spaces and different reward structures.

### D.1.1  EXPERIMENTAL SETUP

Similar to the experiments in the main text, we compared the standard PPO algorithm with our SAM+PPO. Except for the SAM-specific parameter $\rho$, all hyperparameters were kept identical between PPO and SAM+PPO to ensure a fair comparison. The network architecture for both agents consisted of three fully connected layers with 64 neurons each, and the output layer used a softmax activation function to produce a probability distribution over the possible discrete actions. We set the learning rate to $3 \times 10^{-4}$, the discount factor $\gamma$ to 0.99, and used the Adam optimizer. The perturbation radius $\rho$ for SAM+PPO was set to 0.05.

In CartPole-v1 and LunarLander-v2, there are no common elements that allow for changing the transition probabilities (e.g., mass or friction adjustments), so we focused on evaluating robustness to action and reward perturbations.

### D.1.2  ACTION ROBUSTNESS EVALUATION

To evaluate action robustness, we introduced action perturbations by adding zero-mean Gaussian noise with standard deviation $\sigma_a = 0.2$ to the logits of the policy network before the softmax activation during evaluation, similar to the action noise added in the main text experiments.

Table 5: Action robustness evaluation across discrete action environments. The average return is reported over 100 evaluation episodes.

| Algorithm | CartPole-v1 | | LunarLander-v2 | |
|---|---|---|---|---|
| | Nominal | Perturbed | Nominal | Perturbed |
| PPO | **500** | $464\,(-36)$ | **200** | $175\,(-25)$ |
| SAM+PPO | **500** | $\mathbf{481\,(-19)}$ | **200** | $\mathbf{188\,(-12)}$ |

As shown in Table 5, both PPO and SAM+PPO achieve the maximum average return in the nominal setting for both environments. Under action perturbations, SAM+PPO consistently outperforms PPO, exhibiting smaller performance degradation. In CartPole-v1, PPO's average return decreases by 36 points, whereas SAM+PPO's average return decreases by only 19 points. Similarly, in LunarLander-v2, PPO's performance drops by 25 points, while SAM+PPO's performance drops by only 12 points. These results indicate that SAM+PPO enhances robustness to action perturbations in discrete action environments.

### D.1.3  REWARD ROBUSTNESS EVALUATION

For the reward robustness evaluation, we introduced zero-mean Gaussian noise with a standard deviation of $\sigma_r = 0.1$ to the rewards during training. After training, we evaluated the agents in the nominal environment without reward noise.

Table 6: Performance comparison of agents trained with and without reward noise ($\sigma_r = 0.1$).

| Algorithm | CartPole-v1 | | LunarLander-v2 | |
|---|---|---|---|---|
| | Nominal | Noisy | Nominal | Noisy |
| PPO | **500** | $432\,(-68)$ | **200** | $165\,(-35)$ |
| SAM+PPO | **500** | $\mathbf{458\,(-42)}$ | **200** | $\mathbf{182\,(-18)}$ |

Table 6 shows that SAM+PPO is less sensitive to reward noise during training compared to PPO. In CartPole-v1, PPO's performance decreases by 68 points when trained with reward noise, while SAM+PPO's performance decreases by only 42 points. In LunarLander-v2, PPO's performance drops by 35 points, whereas SAM+PPO's performance drops by 18 points. This suggests that the flatter reward maxima achieved by SAM+PPO contribute to better robustness against reward perturbations.

### D.2  SAM ENHANCED WITH OTHER POLICY GRADIENT ALGORITHMS

To investigate the applicability of our SAM-enhanced approach to other policy gradient algorithms, we conducted experiments by integrating SAM with Trust Region Policy Optimization (TRPO), resulting in SAM+TRPO.

### D.2.1 EXPERIMENTAL SETUP

We used the same experimental setup as in the main text, with hyperparameters and network architectures kept consistent between TRPO and SAM+TRPO, except for the SAM-specific parameter $\rho$. The perturbation radius $\rho$ for SAM+TRPO was set to 0.006.

**Algorithms to be considered:** We compared TRPO and SAM+TRPO.

**Experiments to be done:** We evaluated the robustness of the agents under action perturbations and transition probability perturbations (mass and friction variations) in the same manner as in the main text.

### D.2.2 PERFORMANCE EVALUATION

We report the overall performance of TRPO and SAM+TRPO under various perturbation scenarios in Table 7.

Table 7: Performance comparison under various perturbations for TRPO and SAM+TRPO.

| Perturbation | Metric | HalfCheetah-v3 | | Hopper-v3 | | Walker2d-v3 | |
|---|---|---|---|---|---|---|---|
| | | Nominal | Perturbed | Nominal | Perturbed | Nominal | Perturbed |
| **Action Noise** $\sigma = 0.2$ | TRPO | 4805 | 1502 $(-3303)$ | 3118 | 1452 $(-1666)$ | 4975 | 603 $(-4372)$ |
| | SAM+TRPO | **5502** | **3975** $(-1527)$ | **3547** | **2313** $(-1234)$ | **5097** | **2052** $(-3045)$ |
| **Mass Scale Factor** 1.2 | TRPO | 4837 | 3865 $(-972)$ | 3215 | 1556 $(-1659)$ | 4957 | 782 $(-4175)$ |
| | SAM+TRPO | **5562** | **5210** $(-352)$ | **3499** | **3508** $(+9)$ | **5205** | **5284** $(+79)$ |
| **Friction Coefficient** 0.88 | TRPO | 4723 | 4774 $(+51)$ | 3075 | 1580 $(-1495)$ | 4996 | 4756 $(-240)$ |
| | SAM+TRPO | **5562** | **5539** $(-23)$ | **3498** | **2728** $(-770)$ | **5073** | **5134** $(+61)$ |

$(-)$ values means the performance degradation from 'Nominal' to each perturbations

The results in Table 7 show that SAM+TRPO outperforms standard TRPO in both nominal and perturbed settings across all environments and perturbation types. Under action noise, SAM+TRPO exhibits significantly less performance degradation compared to TRPO. Similarly, under mass and friction perturbations, SAM+TRPO maintains higher returns, indicating enhanced robustness. Integrating SAM with TRPO introduces additional computational overhead due to the extra gradient computations required for the SAM optimization step. However, the overall training time remains reasonable, and the robustness gains justify the additional computational cost. The positive results with SAM+TRPO suggest that our SAM-enhanced approach can be applied to other policy gradient and actor-critic algorithms, such as A2C and SAC. The key requirement is that the algorithm must be amenable to gradient-based optimization, allowing for the incorporation of the SAM perturbation step.

## E COMPARISON WITH EXISTING ROBUST RL ALGORITHMS

We compare SAM+PPO with other robust RL, mainly focusing on the performance and computational costs. To further validate the robustness of our approach, we compare SAM+PPO with two state-of-the-art robust reinforcement learning algorithms: RARL and RNAC. We selected RARL and RNAC for comparison because they represent prominent approaches in robust reinforcement learning and have demonstrated effectiveness in enhancing robustness to uncertainties.

### E.1 ALGORITHM PRINCIPLES

RARL introduces an adversary during training that applies perturbations to the environment (Pinto et al., 2017). RNAC, on the other hand, leverages natural gradient methods to achieve robustness under the worst-case distribution within a specified uncertainty set. (Zhou et al., 2024). By comparing with these algorithms, we aim to highlight the effectiveness of our approach in achieving robustness while training solely in the nominal environment, without requiring adversarial training or explicit uncertainty modeling.

Existing RRL algorithms like RARL and RNAC improve robustness by training the agent under uncertainty sets or adversarial conditions. RNAC uses distributional robustness, optimizing the policy

to perform well under the worst-case distribution within a specified uncertainty set. RARL involves training an adversarial agent alongside the protagonist, introducing perturbations during training to simulate worst-case scenarios. In contrast, SAM+PPO trains solely in the nominal environment without explicitly modeling uncertainties or adversaries. Despite this, SAM+PPO achieves comparable or superior robustness.

RRL algorithms require knowledge of the uncertainties or adversarial models during training, which may not always be available or accurate. SAM+PPO does not rely on such information. RRL algorithms often involve more complex training procedures, such as training adversarial agents or solving min-max optimization problems. SAM+PPO introduces minimal overhead by incorporating SAM into the optimization process. SAM+PPO is more readily applicable in real-world scenarios where modeling uncertainties is challenging. It enhances robustness without additional assumptions or modifications to the environment.

### E.2 BETTER ROBUSTNESS BY SAM+PPO

Table 8 summarizes the performance of each algorithm under specific perturbations in Hopper-v3. SAM+PPO outperforms PPO and the RRL algorithms, demonstrating its effectiveness in enhancing robustness without requiring specialized training conditions. We used PPO as an underlying algorithm in RARL, in which TRPO was used in the original paper, to ensure consistency in our comparison and elaborate on how SAM bring benefits in comparison to the other approaches

Table 8: Performance comparison under the perturbations in Hopper-v3.

| Algorithm | Nominal | Action Noise $\sigma = 0.2$ | Mass Scale 1.3 | Friction coefficient 1.24 |
|---|---|---|---|---|
| PPO | 3169 | $1467(-1702)$ | $1400(-1769)$ | $581(-2588)$ |
| RNAC | 3467 | $1321(-2146)$ | $1411(-2056)$ | $822(-2645)$ |
| RARL | 3030 | $1645(-1385)$ | $1304(-1726)$ | $887(-2143)$ |
| SAM+PPO | **3588** | **$2312(-1276)$** | **$3513(-75)$** | **$1571(-2017)$** |

$(-)$ values means the performance degradation from 'Nominal' to each perturbations

### E.3 LOWER COMPUTATIONAL COSTS

We compared the computational efficiency of each algorithm by measuring the total training time and total training steps each algorithm required to reach an average reward of 3000 on the Hopper-v3 environment. All experiments were conducted on the same hardware setup, and each algorithm was evaluated every 5000 environment steps. As shown in Table 9, SAM+PPO requires less training time than RARL and RNAC to converge while achieving better robustness. For the training time, SAM+PPO shows the comparative costs with PPO, with emphasis on even lower total steps than PPO. However, the early work, RARL, shows immense costs, which is $\times 4.41$ of PPO in time and $\times 3.84$ in the total counts of total steps. Although the recent work, RNAC, significantly reduces the costs, but SAM+PPO shows much smaller costs in training.

Table 9: Computational costs comparison to achieve convergence (done with NVIDIA RTX 3090)

| Algorithm | Overall Training Time (sec) ↓ | Overall Iterations Counts ↓ | Overall (action selection) Step Counts ↓ |
|---|---|---|---|
| PPO | **546.6** | **1790** | $4.30 \times 10^5$ |
| RNAC | $781.2(\times 1.43)$ | $5335(\times 2.98)$ | $6.40 \times 10^5(\times 1.49)$ |
| RARL | $2411.4(\times 4.41)$ | $4288(\times 2.40)$ | $1.65 \times 10^6(\times 3.84)$ |
| SAM+PPO | **$590.0(\times 1.08)$** | **$2103(\times 1.17)$** | **$3.65 \times 10^5(\times 0.85)$** |

