# OpenReview forum: "Flat Reward in Policy Parameter Space Implies Robust Reinforcement Learning"
_ICLR.cc/2025/Conference — ICLR 2025 Oral_

### Official Review · Reviewer_jafF · 2024-10-29

**Soundness:** 3
**Presentation:** 3
**Contribution:** 3
**Rating:** 8
**Confidence:** 3

**Summary:**

This work proposes a new method to ensure robustness in RL based on variations in the loss landscape: "SAM": Sharpness-Aware Minimization. By posing the policy optimization as a min/max objective with respect to perturbations in the parameter space, the authors show robustness to changes in reward and dynamics. A theoretical result is given, linking parameter and reward robustness, and a diverse set of experiments on 3 MuJoCo environments is provided.

**Strengths:**

- This use of robustness in policy parameter space seems to be fairly new
- The experiments demonstrate a strong performance boost across a range of perturbations
- The visualizations in Figure 6 offer an interesting insight into the optimizations produced by PPO vs SAM+PPO. The Hopper example is quite striking. Could you elaborate on the distinction and sharp dropoffs seen there?
- Theory provides a potential link between flatness in parameter space and action robustness
- Provided a solid comparison wrt computational overhead / sample complexity and wall time versus other algorithms
- Figure 5 is quite nice, I think it should be emphasized

Overall, the paper seems like a nice first step in the direction of understanding the relationship between robustness in reward, policy parameter, and dynamics spaces. The notion of "flat rewards" is an interesting one.

**Weaknesses:**

**Writing:**

- Overall, I think the clarity of the paper can be enhanced with a re-write fixing grammar and overall structure:
- It would be helpful for example, to also include some visualizations of the definitions 1 & 2.
- Also, Proposition 1 and the following remarks are not very clear. As an example, for Prop1, if I understand correctly, the result would be better phrased as "if $\mathcal{E}$-flat, then $\Delta$-robust, with $\Delta \leq ...$ otherwise the current phrasing is a bit confusing.


**Discussion:**

- The discussion of the main idea, "SAM" is lacking:
- After it is introduced in Sec 3.3, the authors give a way to solve the optimization problem in Eq (3) by their steps (i)-(iv). However, (to me at least), it is not clear why this method is used. Is there prior work demonstrating the efficacy of this method? Are there experiments or maybe some minimal example illustrating the utility of this setup? E.g. why is $\epsilon$ chosen to be in the direction of the previously computed gradient, if theoretically it should represent an arbitrary direction in the ball.
- At the very least, can the authors provide some visual demonstration as to what is happening here in the loss landscape? Getting a better intuition would help to understand the core method of the paper.
- Remark 1.1 seems to be a restatement of Prop 1 unless I am missing something. Could you please explain?
- Remark 1.2 can be improved by using more technically accurate statements (i.e. what is meant by "when a reward function slightly changes")? What is meant by the "direct [correspondence] to the changes of loss function in the supervised learning case"? I think the latter is very unclear, and maybe even misleading.


**Experiments:**
- My only issue with the experiments (minor) is that you are missing RNAC in Table 2 (why?). Also why not compare against RARL? Missing explanation of the shaded regions in each figure caption.

**Questions:**

I'm really curious about "flat rewards" in general. Definitions 1 and 2 seem too strict at first glance (the equalities therein), so it is actually a bit surprising to me that they are even possible at all; however IIUC, Fig 6 does give evidence of this. I think that these definitions can be further elaborated on (do you have a toy example where it is easy to see in parameter or action space?) Realistically, what values of $\epsilon$ do you think are reasonable? Something like $10^{-11}$ or $10^{-2}$? (I might've missed it somewhere, sorry.) If these are novel definitions not previously given in the literature, that can be stated as a contribution of the paper. I think it can spark future work in both theory and experimental directions.

Here are some follow up questions/comments:

- In Sec 5, how long are those agents trained for? Equal number of env steps for each? How were hparams tuned for each algo?
- What is the agent's action scale for these environments (cf L337)? What do you do if the noise added is outside the action range?
- Do you have any ideas about the sharp dropoff in Fig 3b for SAM PPO? it looks interesting, but I'm not sure what to make of it... is there some "critical" mass ratio? I.e., if we zoom in, how sharp is that transition, and have you averaged over enough random seeds?
- you mention "flatter reward maxima" in L70. I think a formal definition or good visualization of this phenomenon early on would really improve the paper.
- How does this work relate at all to other trust region methods like TRPO? How about e.g. [1]

[1]: https://arxiv.org/abs/2103.06257

Typos/minor

- Fig 3 caption "nomial"
- some missing +/- signs in Table 1 (in parens)
- citations in sec 2 often have a missing leading space.
- can you improve the visual in Fig 1? I think it's important but not quite capturing the essence. Maybe just to remove axes and grid and zoom in a bit: is there indeed a channel for the agent? It's hard to see
- The introduction paragraphs have some grammatical issues. A cleanup/re-write here can help to crystallize the main message early on

With a rewrite to clean up the presentation, deeper explanation for SAM (i)-(iv), and perhaps a few more visualizations, this could be a really strong paper; but unfortunately I don't think it's quite there yet.

---

> ### Author Response · Authors · 2024-11-24
> **Official Comment by Authors**
>
> We thank the reviewer for their thoughtful feedback and valuable suggestions. As detailed below, we have addressed the weaknesses and questions raised.
>
> **Writing**
>
> **Weakness 1: Fix grammar and overall structure**
>
> - Thank you for highlighting the need for improved clarity and grammatical correctness. We have undertaken a thorough revision of the paper to address grammatical errors and enhance the overall structure. We have uploaded the revised paper.
>
> **Weakness 2: Visualizations of Definitions 1 & 2**
>
> - We agree that visualizations can greatly aid in understanding formal definitions. In response, we have added illustrative figures to accompany Definitions 1 and 2, in the revised paper.
>
> **Weakness 3: Confused phrasing of  $Δ^∗$-robust and $Δ$-robust**
>
> - We appreciate the reviewer's feedback pointing out the potential confusion in the phrasing of Proposition 1.
> - The notation $\Delta^*$ denotes the specific maximum perturbation magnitude for which the policy $\pi_{\theta^*}$ remains robust, given the $\mathcal{E}$-flatness at $\theta^*$. It is directly derived from the properties of $\theta^*$ and provides a precise bound.
>     - In **Definition 2**, $Δ$-action robustness is defined in general terms, without specifying a particular value for $Δ$.
>     - In **Proposition 1**,  $Δ^∗$ indicates the specific value of $Δ$ that corresponds to the $\mathcal{E}$-flat maximum $θ^∗$.
> - To enhance clarity, we have rephrased the proposition to make the logical implication explicit and to clarify the use of the notation $\Delta^*$.
>     - If $\theta^*$ is an $\mathcal{E}$-flat reward maximum, then the policy $\pi_{\theta^*}$ is $\Delta^*$-action robust, where:
>     - $\Delta^* \leq \||J(\theta^{*})\||\mathcal{E} + \mathcal{O}(\mathcal{E}^2)$

---

> ### Author Response · Authors · 2024-11-24
> **Official Comment by Authors**
>
> **Discussion**
>
> **Weakness 4~6: Main Idea ‘SAM’ is lacking**
>
> - Thank you for pointing out the need for a deeper discussion of SAM (Sharpness-Aware Minimization). We acknowledge that the original explanation may not have sufficiently conveyed the rationale behind the method. we have significantly expanded this section and included detailed explanations in **Appendix C**.
> - We elaborate on the optimization process of the min-max objective in Equation 8, including step-by-step explanations and the inclusion of the pseudocode (Algorithm 1 in Appendix C). This provides clarity on how SAM is integrated with PPO in our approach. Specifically, integrating SAM into PPO requires modifying the standard optimization steps to account for the perturbation $\epsilon$. This involves:
>     - Computing the base gradient $g_{\theta}=\nabla_{\theta}\mathcal{L}(\theta)$ at the parameter $\theta$
>     - Calculating the worst-case perturbation $\epsilon^* = \rho g_{\theta}/||g_{\theta}||$
>     - Performing an additional forward pass to obtain the loss computed at the perturbed parameter: $\mathcal{L}(\theta+\epsilon^*)$
>     - Computing the gradient $g_{\theta^*}=\nabla_{\theta*}\mathcal{L}(\theta^*)$, where $\theta^*=\theta+\epsilon^*$ (at the perturbed parameter)
>     - Updating the policy parameters using the gradient of the perturbed loss, i.e., $\theta \leftarrow \theta - \alpha g_{\theta^*}$, where $\alpha$ is the learning rate.
> - This process introduces additional gradient computations in each optimization step, effectively doubling the number of forward and backward passes compared to standard PPO.
> - **Reason Why Is Chosen in the Direction of the Gradient:** While the inner maximization problem theoretically considers all possible directions within the $ρ$-ball around $θ$, choosing ϵ in the direction of the gradient $g_θ$ is a first-order approximation that captures the worst-case perturbation efficiently. This approximation is rooted in the Taylor expansion of the loss function. The direction of the gradient indicates the direction in which the loss increases most rapidly. By perturbing $θ$ in this direction, we effectively approximate the maximum increase in loss within the allowed perturbation magnitude $ρ$.
> - **Prior Work Demonstrating the Efficacy of The Method SAM:** Foret et al. (2020): The prior SAM-related works have been focused on supervised learning, particularly in computer vision. From the pioneering work of SAM [1], the authors have provided extensive simulation results showing that SAM improves visual classification models in CIFAR-10, CIFAR-100, Flower, ImageNet, etc. It widely proves the efficacy of flat minima in supervised learning. However, it is not the only method to find flat minima. Another work called SWA [2] employs a model-averaging method to find a flatter loss surface, showing that SWA can effectively find flat minima. However, SAM is more widely used than SWA due to its explicit objective function of searching for flat minima. Another learning problem where flat minima works well is the domain generalization of computer vision. As shown in SWAD [3], flat minima are theoretically proven to be a well-generalized domain shift of images, e.g., photograph to sketch, along with outstanding domain generalization performance. For some theoretical studies, [4] further explores the theoretical underpinnings of SAM and demonstrates its effectiveness in different settings. However, we want to point out that the in-depth understanding of flat loss is strongly focused on the image-based supervised learning settings, not on RL society.
> - For the context of RL, as described in the Related Work section [5], a few works have recently reported that the flat reward probably improves the performance of RL. However, the prior work shows the following limitations: It lacks a formal bridge between the flatness in the reward landscape and the robustness in RL, and the effectiveness of flat reward was not carefully examined by varying multiple key perspectives of RL, i.e., action, transition probability, and reward.
> - **SAM visualizations:** We want to say that our visualizations in Fig. 7 are the common way to visualize flatness. In the case of supervised learning (in the original paper of SAM), the difference is that it seeks ‘flat minima’ (our visualizations show a ‘flat maxima’ of rewards). It is hard to provide a conventional visualization of flat loss surface here in the rebuttals. Still, we hope to provide some related papers with standard visualizations of loss function: Fig. 1 in [2] (2d visualization), Fig. 3 in [3] (plot-based visualization).

---

> > ### Comment · Reviewer_jafF · 2024-11-25
> > **Response to first two comments**
> >
> > Thanks a lot for addressing my comments in detail. I am taking some time to read and digest your comments to the other reviewers as well. I think your responses and the re-written version of the paper are quite good.
> >
> > Here are some quick comments about the new Appendix C. Thank you for adding these details, but I still have some confusion:
> >
> > - Eq 28 is confusing - this is a one-shot SL problem, whereas the true PPO loss function is concerned with entire trajectories (as you write in Alg1, p18). It seems like you hint at this mismatch in the next line, "which can be extended to Equation 8 in Section 5, by..." I might be confused but it seems like more than "extending", I think Eq28 is just not valid for the temporal setting in RL (it is equivalent to setting $\gamma=0$).
> >
> > - Also after looking at https://arxiv.org/pdf/1901.09184 it seems to me that they consider the full discounted return, not one step, as you wrote in Eq. 2. Could you please clarify this for me?
> >
> > - You then mention "corresponding proof in Section 4 of the main text", but I don't see any proof there. Can you elaborate or maybe fix the typo?
> >
> > - Missing "." on L902
> >
> >
> > I appreciate your step-by-step description of the algorithm. It helped me understand my previous concern about choosing the direction in this $\rho$-ball, and now seems like a well-founded idea. (I also should have looked more carefully at the original SAM work: https://arxiv.org/pdf/2010.01412 )
> >
> > - As you mention, based on Eq6 and 7 in App. C; it seems like SAM+X needs 2x as many gradient steps and 2x as many forward passes. Assuming the environment step speed is negligible in comparison, does this mean SAM+X takes roughly 2x as much wall time? Is there a way to estimate either of these values without explicit re-calculation? Can you use the trick used in the original SAM paper? If possible, it could greatly improve the efficiency!

---

> ### Author Response · Authors · 2024-11-24
>
> **Weakness 7: Remark 1.1 and Proposition 1**
>
> - Proposition 1 establishes a theoretical link between reward surface flatness and robustness to action perturbations in reinforcement learning. Specifically, it states that if a policy is $\mathcal{E}$-flat in the parameter space, then it is  $\Delta^*$-robust with respect to action perturbations, where $\delta ≤$  $\Delta^*$. This proposition provides formal linking that **flatness in the policy parameters leads to robustness against the perturbations in actions**, as it ensures that perturbations in parameters (and consequently, actions) do not significantly affect the expected reward.
> - Remark 1.1 serves to highlight the practical significance of the theoretical result. It explains that the SAM optimization method we employ leads to the action robust MDP objective. This provides a clear justification for using SAM in our approach to achieve robustness to action perturbations.
> - Remark 1.1 extends the theoretical result of Proposition 1 by highlighting its practical application in our approach. It emphasizes that by applying the SAM optimization (Equation 3) to the standard reinforcement learning objective, we effectively obtain the action robust MDP objective (Equation 2). The remark underscores that the optimization process introduced by SAM aligns with the robust optimization framework of action robust MDPs. Specifically, optimizing the SAM objective inherently addresses the robustness to action perturbations as formalized in robust MDP formulations.
>
> **Weakness 8: Remark 1.2**
>
> - We revised the statements in Remark 1.2 by using technically accurate statements. The revised statements are as follows.
>     - (L303)‘For the changes of reward function’ → ’For reward function perturbations’
>     - (L304) ‘direct [correspondence] to the changes of loss function in the supervised learning case’ → ‘it directly corresponds to the perturbations of loss function in the supervised learning case’
>     - (L305) ‘when a reward function slightly changes’ → ‘when a reward function has merely slight perturbations’
>     - (L306)When the MDP’s transition probability changes → When the MDP’s transition probability has perturbations
> - **Clarifying the Analogy to Supervised Learning:** **In supervised learning, robustness to changes in the loss function (e.g., from label noise or adversarial examples) is achieved when the model's parameters are in flat regions of the loss landscape. This means that small perturbations in the inputs or outputs lead to minimal changes in the loss. Similarly, in reinforcement learning, robustness to changes in the reward function or transition dynamics can be achieved when the policy parameters lie in flat regions of the expected reward landscape. By promoting flatness in the reward landscape, we make the policy less sensitive to small perturbations in the environment, whether they arise from changes in rewards or transitions.
>
> [1] Foret, P., Kleiner, A., Mobahi, H., & Neyshabur, B. (2020). *Sharpness-Aware Minimization for Efficiently Improving Generalization*. International Conference on Learning Representations (ICLR).
>
> [2] Izmailov, P., Podoprikhin, D., Garipov, T., Vetrov, D., & Wilson, A. G. (2018). Averaging weights leads to wider optima and better generalization. *arXiv preprint arXiv:1803.05407*.
>
> [3] Cha, J., Chun, S., Lee, K., Cho, H. C., Park, S., Lee, Y., & Park, S. (2021). Swad: Domain generalization by seeking flat minima. (Neurips)
>
> [4] Zhuang, L., Niu, G., & Sugiyama, M. (2021). *Surrogate Gap Minimization Improves Sharpness-Aware Training* International Conference on Learning Representations (ICLR)
>
> [5] Sullivan, R., Terry, J. K., Black, B., & Dickerson, J. P. (2022, June). Cliff Diving: Exploring Reward Surfaces in Reinforcement Learning Environments. (ICML)
>
> **Experiments**
>
> **Weakness 9: Missing result in Table 2**
>
> - We have additionally included the result of RNAC and RARL in the Table 2
> - The table below shows the performance comparison of agents trained with and without reward noise ($σ_r$ = 0.1).
> - SAM+PPO outperforms RNAC and RARL in the reward noisy cases.
>
> | Algorithm | HalfCheetah-v3 |  | Hopper-v3 |  | Walker2d-v3 |  |
> | --- | --- | --- | --- | --- | --- | --- |
> |  | Nominal | Noisy | Nominal | Noisy | Nominal | Noisy |
> | PPO | 4820  | 3688(−1132)  | 3150  | 2945(−205)  | 4780  | 2204(−2576) |
> | RNAC | 5423  | 4088(−1335)  | 3211  | 3035(−176)  | 4184  | 3172(−1012) |
> | RARL | 5620  | 4617(−1003)  | 3124  | 2993(−131)  | 4388  | 3085(−1303) |
> | SAM+PPO | **6530**  | **5990(−540)**  | **3505**  | **3377(−128)**  | **5120**  | **4226(−894)** |

---

> > ### Comment · Reviewer_jafF · 2024-11-25
> > **Response to third comment**
> >
> > Thanks for giving some discussion on Prop 1 vs Remark 1.1. Can you please see my earlier concerns about the reward vs return objective? I see again in Eq7 that you're maximizing just the one step reward. Could you elaborate a bit on that? Thanks for clarifying Rmk 1.2, it makes more sense and I have a better mental picture of it now.
> >
> > I think adding RNAC, RARL, and even SAM+TRPO improves the experimental nature of the paper.
> >
> > Can you please comment on how there could be positive reward shifts when perturbations are present (e.g. in Table 1)? Is this statistically significant? How many runs were performed, and what is the standard deviation? Nevertheless, the addition of the other algorithms here helps gain a better understanding of the broader picture.
> >
> >
> > **Minor Ref Issues**:
> >
> > There are still missing spaces in front of citations in Sect. 2.
> >
> > Also, I noticed a few refs were out of date (arxiv instead of published versions), e.g. "Aravind Rajeswaran, Sarvjeet Ghotra, Balaraman Ravindran, and Sergey Levine. Epopt: Learning
> > robust neural network policies using model ensembles. arXiv preprint arXiv:1610.01283, 2016" is in ICLR 2017.
> >
> > "On large-batch training for deep learning: Generalization gap and sharp minima" is also in ICLR 2017.
> >
> > Same for Ota et al, here's an up to date ref:
> > @article{ota2024aframework,
> >   author = {Ota, Kei and Jha, Devesh K. and Kanezaki, Asako},
> >   title = {A Framework for Training Larger Networks for Deep Reinforcement Learning},
> >   journal = {Machine Learning},
> >   year = {2024},
> >   month = jun,
> >   day = {05},
> >   issn = {1573-0565},
> >   doi = {10.1007/s10994-024-06547-6},
> >   url = {https://doi.org/10.1007/s10994-024-06547-6},
> > }
> >
> > The Sutton & Barto book citation seems off.

---

> > > ### Comment · Reviewer_jafF · 2024-11-25
> > > **minor comments**
> > >
> > > I would highly suggest combining Table 1 with Table 7 to get a better picture. It is also good to show that adding SAM to TRPO improves its performance, which I believe deserves attention in the main text.
> > >
> > > Can you explain the bolding in Table 1? I'm confused about the choice for Hopper (Action Noise).
> > >
> > > Also, Fig 1 is looking better now. Purely for aesthetics, removing the axis ticks and labels, and maybe even combining both into one bigger figure can help even more - just my opinion.
> > >
> > > Also, thanks for the additional illustrations, it helps give a better quick intuition.

---

> > > > ### Comment · Reviewer_jafF · 2024-11-25
> > > > **Proposition 1**
> > > >
> > > > After reading the other reviews and seeing that Proposition 1 (and its proof) have changed a bit - it raises the following concern:
> > > >
> > > > You've provided only an upper bound on $\Delta^*$, meaning that in principle $\Delta^* \to 0$ is possible. This would make the following Remark vacuous! If $\Delta^* \to 0$, then the resulting policy is *not* action robust.
> > > >
> > > > Thus, it seems like instead of an upper bound, we really need a lower bound on $\Delta^*$. Given (A) the limited amount of time, (B) I did not bring this up in my original review and (C) I'm viewing the paper as mostly experimental at this point; I would conclude that this is not entirely detrimental. However, it is a bit concerning from a theoretical standpoint.
> > > >
> > > > If you can provide some hints on how to proceed for future work or reduce the theoretical claims (esp. in connection with my previous concerns about reward vs trajectory return), I think that would be the most appropriate way to proceed.
> > > >
> > > > I look forward to hearing back from the authors!

---

> > > > > ### Author Response · Authors · 2024-11-27
> > > > >
> > > > > We sincerely appreciate your careful and thorough feedback to improve our work further. Also, thank you for waiting for our response. Within the limited rebuttal period, we have tried our best to fully address your concerns and revise the manuscript to accept your constructive suggestions.
> > > > >
> > > > > **Comment 1, 2, 6: Revision for the form of cumulative reward**
> > > > >
> > > > > - Fortunately, by accepting your careful comments, we have revised the manuscript to represent the loss function of PPO in the form of cumulative rewards.
> > > > > - Starting by the confusion from Eq. (28), there are many related parts to the revision for the cumulative reward form. Specifically, we have used the revised formulation, i.e.,$\max_{\pi} \min_{\|\delta_t\| \leq \beta} \mathbb{E}{s \sim p, a \sim \pi} \left[ \sum_{t=0}^\infty \gamma^t r(s_t, a_t + \delta_t) \right]$ , in the object function of Action robust MDP (Equation 2. in Section 3.). Also, by following the change, Definition 1,2, remark 1.1, 1,2, and the proof of Proposition 1 (in Appendix A) have been revised. Following are the corresponding revisions and their parts:
> > > > > - Equation 2 : $\max_{\pi} \min_{\|\delta_t\| \leq \beta} \mathbb{E} {p, \pi} \left[ \sum_{t=0}^\infty \gamma^t r(s_t, a_t + \delta_t) \right]$
> > > > > - Definition 1 (Equation 4) : $\quad \mathbb{E}{s{\sim p}, a \sim \pi_{\theta^{*}+\epsilon}{(a|s)}} {\left[ \sum_{t=0}^\infty \gamma^t r(s_t,a_t) \right]}$
> > > > > - Definition 2 (Equation 5) : $\mathbb{E}{s\sim p,a\sim\pi_{\theta^{*}}}{\left[ \sum_{t=0}^\infty \gamma^t r(s_t,a_t + \delta_t) \right]}$
> > > > > - Remark 1.1 (Equation 7) : $\mathbb{E}{s \sim p, a \sim \pi_{\theta}}{\left[ \sum\textstyle_{t=0}^\infty \gamma^t r(s_t,a_t + \delta_t) \right]}$
> > > > > - Appendix A
> > > > >     - $\mathbb{E}{s \sim p, a \sim \pi_{\theta^* + \epsilon}} \left[ \sum_{t=0}^\infty \gamma^t r(s_t, a_t) \right] = r^*.$
> > > > >     - $\left| \mathbb{E}{s, a} \left[ \sum_{t=0}^\infty \gamma^t  (r(s_t, a_t + \delta_t) - r(s_t, a_t)) \right] \right| \leq \frac{L_r  (\| J(\theta^*) \| \mathcal{E} + \mathcal{O}(\mathcal{E}^2))}{1 - \gamma}$
> > > > > - When $\gamma=0$, as you pointed out, it becomes the previous version of the equation.
> > > > > - We believe that the change to cumulative reward strictly corresponds to the actual policy training with the cumulative reward. We sincerely appreciate your careful suggestions.
> > > > >
> > > > > **Comment 3: Inappropriate reference to the proof**
> > > > >
> > > > > - Thank you for pointing out the our mistake. We have revised the line as below, clearly stating that Proposition 1 and Remark 1.1 are from Section 4 of the main text, and the corresponding proof is from Appendix A.
> > > > >     - (Before revision: L899) ‘Based on Proposition 1, Remark 1.1, and the corresponding proof in Section 4 of the main text,’
> > > > >     - (After revision: L911) ‘Based on Proposition 1, Remark 1.1 in Section 4 of the main text, and the corresponding proof in Appendix A,’
> > > > >
> > > > >
> > > > > **Comment 4: Missing “.” on line 902**
> > > > >
> > > > > - We added missing “.” on line 914 of the revised manuscript
> > > > >
> > > > > **Comment 5 : Computational overhead and possible efficient tricks of applying SAM**
> > > > >
> > > > > - To the best of our understanding, you refer “the trick” from the original SAM paper, as the approximation of SAM gradient via first-order estimation (referring Eq. 3 in the original SAM paper). The straightforward computation of SAM gradient requires Hessian computation, but it can be relieved by using “the trick”, which uses two times gradient computations (at $\theta$, and at perturbed $\theta$). The trick is exactly what we used in our evaluations. Also, our algorithmic description is also based on the trick.
> > > > > - (If you are not referring the trick described above, please let us know! Until end of the discussion period, we will try to do our best for providing meaningful results)
> > > > > - Also, the efficient method is widely-used in SAM-related researches, because the naive computation of Hessian hinders massive experiments of deep models.
> > > > > - **Consequently, our complexity analysis is already based on the efficient version of SAM.**
> > > > > - We have found other related methods in public implementations as alternative ways to compute SAM gradient, but we do not use it due to the lack of peer-reviewed level of reliability.

---

> > > > > ### Author Response · Authors · 2024-11-27
> > > > >
> > > > > **Comment 7: How could there be positive reward shifts in perturbed evaluation?**
> > > > >
> > > > > - First of all, we have performed 100 evaluation runs to compute the average performance. We conjecture that the number of trials is quite large to trust the mean performance.
> > > > > - However, standard deviations are quite large in our evaluations (as shown in the colored interval of each plot in the figures). Also, we want to point out that RL environments commonly show fairly larger variances than the conventional classification testing (in accuracy).
> > > > > - Also, the observed ‘positive reward shifts’ are indeed much smaller than the reward performance values, i.e., about tens (positive shift) for thousands (performance) in values.
> > > > > - By keeping these in our mind, we conclude that **i)** The mean performance is statistically reliable, **ii)** Some cases with positive shifts do not meaningful gains when seeing large reward performance and variances.
> > > > > - However, from a high-level viewpoint, this kind of minimal degradations or even a slight gain can occur when the changes in environment is not challenging to agents. As widely known, deep models are naturally capable to generalize on unseen samples/tasks/environments, thus such outliers with a small positive shifts can be possible.
> > > > >
> > > > > **Comment 8: Minor Ref Issues**
> > > > >
> > > > > - Thank you for providing details with Section 2 and references. We reviewed the whole manuscript for missing spaces before citations and identified any outdated references
> > > > > - Update outdated references:
> > > > >     - Rajeswaran, A., Ghotra, S., Ravindran, B., & Levine, S. (2017). EPOpt: Learning Robust Neural Network Policies Using Model Ensembles. In *Proceedings of the International Conference on Learning Representations (ICLR)*.
> > > > >     - Keskar, N. S., Mudigere, D., Nocedal, J., Smelyanskiy, M., & Tang, P. T. P. (2017). On Large-Batch Training for Deep Learning: Generalization Gap and Sharp Minima. In *Proceedings of the International Conference on Learning Representations (ICLR)*.
> > > > >     - Ota, K., Jha, D. K., & Kanezaki, A. (2024). A Framework for Training Larger Networks for Deep Reinforcement Learning. *Machine Learning*. Advance online publication. https://doi.org/10.1007/s10994-024-06547-6
> > > > >     - Sutton, R. S., & Barto, A. G. (2018). *Reinforcement Learning: An Introduction* (2nd ed.). MIT Press.
> > > > > - We sincerely appreciate your careful comments.
> > > > >
> > > > > **Comment 9: Combining Table 1 and Table 7**
> > > > >
> > > > > - We appreciate your thoughtful suggestion on the experiments to be more comprehensive.
> > > > > - We also think the combined table will be more comprehensive. However, due to the strict page limit of 10 pages, we were unable to include the combined table in the main text without exceeding the allowed length. Instead, we added a line for clearly indicating that the additional experiment of applying SAM on other RL algorithm is presented in the appendix.
> > > > >     - (L332) ‘Additional evaluations of other RL algorithm and SAM enhanced version are presented on Appendix D.2’
> > > > >
> > > > >
> > > > > **Comment 10: Bold font in Table 1**
> > > > >
> > > > > - We used boldface type fonts in Table 1 to highlight two key aspects for each environment.
> > > > >     - Highest performance on ‘Nominal’ result,
> > > > >     - Smallest performance degradation on ‘Perturbed’ result.
> > > > > - To improve clarity, we have included a brief explanation in the text accompanying the table to describe how the highlighting is applied
> > > > >     - (L463) ‘Bold face is used for Highest performance in ’Nominal’, smallest performance degradation in ’Perturbed’’
> > > > > - We have corrected the usage of boldface fonts in Table 1.
> > > > >
> > > > > **Comment 11: Revising Figure 1**
> > > > >
> > > > > - Thank you for suggestions to enhance the visual presentation on Figure 1.
> > > > > - We removed axis ticks and labels in the figure, as you mentioned. We have plotted the two figures in a single, large figure to provide direct comparison, but found that the intention to highlight the narrow path for SAM+PPO avoid, be reduced.
> > > > > - Although maintaining the two figures separately, we made Figure 1 more clear and interpretable.

---

> ### Author Response · Authors · 2024-11-24
>
> **Question 0: Discussions of the range of flatness in reward surface**
>
> - When going back to the pioneering work of SAM, it is infeasible to anticipate how much a wide region of flatness around the solution can be achieved after training deep architecture via SAM. Herein, we tried to do our best to provide a general intuition for thinking of the range of flatness.
> - For the SAM objective, $\rho$ works as a critical hyperparameter to control the range of flatness. It is the radius of allowed perturbations on parameter space. Because SAM finds the worst cases within the radius $\rho$ around the parameter, it forces the RL agent to search for the policy, where it shows minimal reward decrease within a sphere of radius $\rho$ around the policy parameter. Thus, ideally, sufficiently trained SAM+PPO would show $\rho$-radius flat reward region.
> - In practice, we used $\rho=0.01$, thus it can be interpreted that SAM+PPO aims to find a 0.01-radius flat reward surface. However, it does not mean that a larger $\rho$ is beneficial. Too large radius probably makes the policy struggle to find too-wide flat maxima, which is rarely existing in the parameter space. Therefore, it is crucial to find the optimal $\rho$. We have added the ablations in Appendix C.3.
> - We want to point out that this intuition is not solely new to our work but is widely accepted in the prior SAM-related literature. However, the prior literature focuses on supervised learning, not reinforcement learning.
>
> **Question 1: Training steps, hyperparameters**
>
> - All agents were trained for 3,000,000 environment steps. We ensured that each agent had the same amount of interaction with the environment by maintaining an equal number of environment steps. This approach provides a fair comparison of performance and learning efficiency among the different algorithms.
> - For PPO, SAM+PPO, and RNAC experiments, we adopted the hyperparameters provided in the RNAC paper for each environment. This includes settings such as learning rates, batch sizes, discount factors, GAE lambda, and other algorithm-specific parameters. For RARL, we used the hyperparameters specified in the RARL paper. The only hyperparameters we tuned were those introduced when applying SAM to PPO, specifically the SAM perturbation radius ($ρ$). We experimented with different values of $ρ$ (e.g., 0.01, 0.05, 0.1) to find the optimal setting that enhances robustness without adversely affecting training stability. The complete list of hyperparameters for each algorithm and environment is additionally provided in Appendix B.2.
> - By using the hyperparameters from the RNAC and RARL papers, we aimed to ensure that our experiments are directly comparable to prior work and that any performance differences are due to the algorithms themselves rather than differing hyperparameter choices.
>
> **Question 2: Agent's action scale, dealing with noise added outside of the action range**
>
> - In the environments we used, the action spaces are continuous and bounded. For example, in MuJoCo environments like Hopper-v3 and Walker2d-v3, the action values are within the range [-1, 1] for each action dimension.
> - When evaluating robustness to action perturbations, we add Gaussian noise to the agent's actions during testing. If the addition of noise results in actions outside the valid range, we clip the actions to the allowable bounds of the environment. This ensures that all actions remain valid and prevents errors during simulation. We have clarified this procedure in Section 5.2, specifying how action noise is handled and how actions are kept within the valid range.

---

> ### Author Response · Authors · 2024-11-24
>
> **Question 3: Ideas of sharp performance dropoff in Hopper-v3 of high mass factor (Figure 4b) for SAM+PPO**
>
> - Thank you for your insightful observation regarding the sharp drop-off in performance for SAM+PPO in Figure 4b. We have investigated this phenomenon and would like to explain.
> - The sharp decline in performance at higher mass coefficients, particularly noticeable at a mass coefficient of 1.4 in SAM+PPO(also 1.3 in RARL, and both 1.1 in RNAC, PPO), is due to the physical limitations inherent in the Hopper environment. Hopper is designed with a single leg and relies on precise balance and sufficient torque to propel itself forward and maintain stability.
> - As the mass coefficient increases, the agent becomes significantly heavier while the actuator limits (maximum torque outputs) remain unchanged. There is a critical mass ratio beyond which the actuators cannot generate enough force to counteract the increased gravitational force on the heavier body. This results in the robot failing to make forward progress or maintain an upright position, leading to episodes terminating prematurely due to falls.
> - The critical mass ratio is influenced by both the environmental constraints and the algorithm's ability to adapt to those constraints.
> - We added the performance evaluation of RARL in the main experiment and found that RARL also exhibits the sharp drop-off performance of a high mass coefficient. Up to this point, SAM+PPO and  RARL can adapt to the changes in mass by adjusting their control policies, thanks to their robustness mechanisms. However, beyond this critical mass, the task becomes physically infeasible for the agent to perform, regardless of the robustness of the policy.
> - Also, the performance was averaged over 100 evaluation episodes for each mass coefficient value, providing statistically significant results and confirming the reliability of the observed phenomenon.
>
> **Question 4: Definition of flat reward maxima and visualization of the phenomenon**
>
> - We defined  $\mathcal{E}$-flat reward maxima in Section 4, along with the preliminaries needed to understand the definition. Also, We added the illustrative figures of the definition $\mathcal{E}$-flat reward maxima and  $\Delta$-action robustness (referring to Figure. 2 in the revised paper).
>
> **Question 5: How does SAM relate to other methods like TRPO**
>
> - Both SAM+PPO and TRPO aim to improve policy optimization by controlling the update step to promote stability and robustness. While TRPO enforces a trust region via constraints on the KL divergence, SAM promotes flatness in the loss landscape through parameter perturbations. We have extended our work to integrate SAM with TRPO, creating SAM+TRPO. We have included experimental results comparing SAM+TRPO with standard TRPO and other baselines. The results, presented in Appendix D, show that SAM can enhance TRPO's performance by further promoting robustness.
>
> | Perturbation | Metric | HalfCheetah-v3 |  | Hopper-v3 |  | Walker2d-v3 |  |
> | --- | --- | --- | --- | --- | --- | --- | --- |
> |  |  | Nominal | Perturbed | Nominal | Perturbed | Nominal | Perturbed |
> | Action Noise σ = 0.2 | TRPO | 4805 | 1502 (−3303) | 3118 | 1452 (−1666) | 4975 | 603 (−4372) |
> |  | SAM+TRPO | **5502** | **3975 (−1527)** | **3547** | **2313 (−1234)** | **5097** | **2052 (−3045)** |
> | Mass Scale Factor 1.2 | TRPO | 4837 | 3865 (−972) | 3215 | 1556 (−1659) | 4957 | 782 (−4175) |
> |  | SAM+TRPO | **5562** | **5210 (−352)** | **3499** | **3508 (+9)** | **5205** | **5284 (+79)** |
> | Friction Coefficient 0.88 | TRPO | 4723 | **4774 (+51)** | 3075 | 1580 (−1495) | 4996 | 4756 (−240) |
> |  | SAM+TRPO | **5562** | 5539 (−23) | **3498** | **2728 (−770)** | **5073** | **5134 (+61)** |
>
> **Question 6: Typos and minor issues**
>
> - Thank you for catching typos and We appreciate your attention to detail. We have corrected Figure 4’s caption, reviewed Table 1 and added the missing "+/-" signs, corrected the citation formatting to include the necessary leading spaces. Also we have thoroughly revised the introduction to address grammatical issues and improve the overall clarity.
> - We have redesigned Figure 1 by adding the mini-figure that has increased the zoom level, to make key features more visible, illustrating that there indeed is a narrow path for the agent.
>
> We truly appreciate your time for reviewing our work. We hope our response clarifies the raised concerns.

---

> ### Author Response · Authors · 2024-11-27
>
> **Comment 12: Future plan for improving theoretical claims** (lower bound on $\Delta^*$)
>
> - As you pointed out, we acknowledge that the formulation of the lower bound of the robustness would be the ultimate goal. If possible, by finding the lower and upper bounds, we would fully understand how much robustness in action can be achieved by using the flatness on parameter space.
> - **First, we here emphasize the meaning of upper bound that achieved in our work.**
> - The key point in the Proposition is that $\Delta^*$ represents the maximum allowable action perturbation under which the policy $π_{θ^{∗}}$ maintains its optimal expected cumulative reward, the largest perturbation magnitude for which the policy remains robust.
> - Addressing concern about $\Delta^*$ → 0: Let’s consider the components $\Delta^*$ :
>     - $\mathcal{E}$ is positive by Definition 1. If $\mathcal{E}$ = 0, implying that reward is not flat, which contradicts to the flat reward maxima. We assume the given flat reward on the parameter space, thus $\mathcal{E}>0$.
>     - $||J(θ^∗)∥$ is non-negative unless the policy is completely insensitive to parameter change. If $||J(θ^∗)∥=0$, the policy is constant with respect to $\theta$. We want to point out that the Jacobian is the derivative of policy with respect to $\theta$, not a derivative of reward (or loss). Therefore, even on reward maxima, the Jacobian does not have to be zero-forced.
>     - For the higher-order term of $\mathcal{O}(\mathcal{E}^2)$, the term is related to the 2nd and higher order of derivatives of policy with respect to $\theta$. Thus, a zero value for the term indicates that the policy is fixed to be constant w.r.t. $\theta$; deep policy model outputs same value even with the changes in parameter space. Such cases do not happen in a practical training.
>     - Therefore, we believe that the case of $\Delta^*$ → 0 would not happen.
> - **Let us show our insight on achieving the lower and upper bounds, possible done in the future work.**
> - The problem is to find how the variations in parameter space affects to the variations in action space; bridging the parameter variations to the output variation of policy network. With some further formula, it is a problem of converting a sphere around $\theta^*$ (flat reward region in parameter space) into the corresponding closed region of action space around $a$ (a clean region) who keeps the reward be flat.
> - Unfortunately, it is intractable to exactly formulate the region on the action space for a given sphere in parameter space. For a starting point of the analysis in the future work, when assuming a linear model, we expect to find the exact solution of the flat region on the action space (no needs for bounds, but giving exact solution). Briefly, it would say that *“for a flat reward region on the parameter space, there exists a action region where all possible actions in the region give flat reward”.*
> - As a further direction for deep models, we guess that the exact solution of the action region is intractable, but we imagine that a subset and superset can be found. By saying the subset and superset, we expect to provide the lower and upper bound of the flatness on the action space. We think that our Proposition 1 would be a preliminary form of the upper bound.
>
> - To our opinion, we think that our theoretical claims are not perfect, but show valuable insights for linking between flat reward in parameter to action robustness. We greatly thank that the claims are indeed improved by your suggests for considering the cumulative reward form.
> - We appreciate your insightful feedback and we hope that this clarification alleviates your concerns.
>
> Thank you once again for taking the time to review our work. We hope our response clarifies the raised concerns.

---

> > ### Comment · Reviewer_jafF · 2024-11-27
> > **Response to reviewers**
> >
> > Thank you again for such an in-depth response. This set of responses, as well as those to other reviewers have changed my mind a bit. I stand by my original view that this is an interesting first step in SAM-style ideas applied to RL, which I believe will open avenues for future theoretical and experimental contributions. However, the authors have now significantly improved the manuscript in *three* regards (1) experimental justification with additional baselines, (2) theoretical justification with expanded discussion and clarification (the "reward to return" fix was crucial!) and (3) an overall improvement in writing and presentation quality. Therefore, I am happy to recommend acceptance of this work and I look forward to seeing how it impacts the field.

---

> > > ### Author Response · Authors · 2024-11-27
> > >
> > > Thank you again for your thoughtful and constructive feedback. We are encouraged to hear that our responses and the subsequent revisions have positively influenced your perception of our work. Your insightful review has been instrumental in guiding the significant improvements made to our manuscript.We look forward to the opportunity to contribute further to the field and to see how our work will influence future research.

---

### Official Review · Reviewer_uQqo · 2024-11-01

**Soundness:** 2
**Presentation:** 2
**Contribution:** 2
**Rating:** 6
**Confidence:** 3

**Summary:**

This paper explores the impact of flat minima in reinforcement learning (RL), linking flatter reward surfaces to improved model robustness. The authors show that flatter rewards lead to more consistent actions despite parameter changes, enhancing robustness against variations in state transitions and reward functions. The authors show through extensive experiments to confirm that flatter rewards significantly bolster RL model performance across diverse scenarios.

-------------------
After the rebuttal: The authors have addressed some of my concerns. I raised the score.

**Strengths:**

- Provide a link of flat reward to action robustness. The authors show this through both theoretical results in section 4, and various experiment results. The motivation of having a robust objective is good. The theoretical result seems correct.

- Positive experiment results showing the benefit of optimizing for a flat reward maxima. The authors show this through different experiment settings: variation to physics properties of the underlying MDP, and visualization of the reward surface.

**Weaknesses:**

- The performance of SAM + PPO is mixed in comparisons to the baselines, e.g. some visible ones at Fig 5.c, 4.b.

- Ablations are not provided to understand how such an objective can bring benefits in comparisons to similar approaches, e.g. RNAC or robust RL.

**Questions:**

- Is the perturbation domain $\rho$ in Eq.8 known to the agent? Probably the optimization of the objective in Eq.8 needs elaboration, and with pseudo-code.

- Why in "Nominal"  SMA+PPO still has a higher reward, e.g. Table 1+2, Fig. 3,4. Similarly,  experiment in 5.2, why when action noise is small, i.e. even equal to 0, SAM+PPO still performs better than the others, because the objectives of PPO and SAM+PPO would converge to the same one? And in 5.3, SAM+PPO has a higher return, while with variation in Friction Coefficient shows mixed results.

- Joint variation of friction and mass shows quite clear that SAM+PPO is performing better than baselines, except on Walker2d-v3 with a mixed result. Can the authors elaborate on why or provide ablation to explain the mixed performance of SAM+PPO?


- The proof of proposition 1 is a bit not standard. The policy is sometimes referred as a distribution, but sometime used as a deterministic mapping. It needs revised.

---

> ### Author Response · Authors · 2024-11-24
> **Official Comment by Authors**
>
> We thank the reviewer for their thoughtful feedback and valuable suggestions. As detailed below, we have addressed the weaknesses and questions raised.
>
> **Weakness 1 and Questions 2, 3: Mixed evaluation results**
>
> - **Questions of “Why in "Nominal" SAM+PPO still has a higher reward, e.g. Table 1+2, Fig. 4,5. Similarly, experiment in 5.2, why when action noise is small, i.e. even equal to 0, SAM+PPO still performs better than the others, because the objectives of PPO and SAM+PPO would converge to the same one?”:** To the best of our understanding, the issue is about the better performance of SAM+PPO over PPO, even in a nominal case without perturbations. To answer this, we want to refer to the fact that SAM is also effective in improving the performance of models even without train-test discrepancy. For example, the original paper of SAM [1] and the related works [2] (Stochastic Weight Averaging (SWA); another approach to find flatter minima) widely validate that the flatter minima improve the model performance in popular benchmarks, including CIFAR-10, CIFAR-100, Flower, ImageNet, etc. It is because the train and test splits are commonly separated from each other, so the deep models need to be well-generalized to test splits, which is distinct from the train splits. For image classifications, the train and test images are different even with the same categories. For reinforcement learning, we separately generate different test episodes even with the same configurations of environments. Due to the gap between train and test splits, deep models commonly suffer from performance degradation in testing. **In this point of view, flatter minima via SAM is widely accepted to improve the generalization performance in testing. That is why SAM+PPO outperforms PPO in some nominal cases.** Also, even in training with the nominal case, **the minima found by SAM+PPO are surely different from the minima by PPO**. We want to remind you that SAM+PPO considers the flatness of loss surfaces, but PPO does not. Therefore, PPO can find sharper minima, but SAM+PPO is forced to find flatter minima.
>
> [1] Foret, P., Kleiner, A., Mobahi, H., & Neyshabur, B. (2020). *Sharpness-Aware Minimization for Efficiently Improving Generalization*. International Conference on Learning Representations (ICLR).
>
> [2] Izmailov, P., Podoprikhin, D., Garipov, T., Vetrov, D., & Wilson, A. G. (2018). Averaging weights leads to wider optima and better generalization. *arXiv preprint arXiv:1803.05407*.
>
> - **Further explanation on high friction cases:** As the reviewer acknowledged, SAM+PPO shows clear performance gains in the joint variations of friction and mass. However, as pointed out, in the Walker2d-v3 case, SAM+PPO outperforms others in low frictions but not in high frictions (quite mixed results).
> - It is challenging to fully understand the empirical results, particularly in specific settings, i.e., the high friction regime. However, we have tried our best to explain the results as follows:
> - First of all, compared with HalfCheetah-v3 and Hopper-v3, the Walker2d-v3 environment is particularly sensitive to changes in friction due to the complex dynamics of balancing and coordinating two legs.
> - Second, the increasing friction from low to high can be understood as the dramatic changes in environments. The agent of Walker2d-v3 has its center of mass at the upper side of its body compared with HalfCheetah-v3 and Hopper-v3. Thus, the high friction makes the agent easily fall over.
> - Third, SAM does not directly perturb actions but indirectly perturbs actions via perturbing the policy parameters. So, SAM can experience as various as actions that might be found by perturbing policy parameters. In contrast, other action-robust RL like RARL explicitly perturbs the actions to experience the worst case.
> - Based on the aspects above, we conjecture that SAM+PPO focuses less on experiencing high friction cases, which are dramatic changes in environments.
> - However, we want to point out that SAM+PPO can widely experience actions, transitional probability, and reward shifting by “indirectly” perturbing the policy parameters. As empirical evidence, SAM+PPO outperforms PPO in the various factors of perturbations, but RNAC or RARL shows comparably limited gains over PPO (referring to Figure 6 and Table 1 in the main paper).
>
> To our understanding, we here provide the following additional results and discussions for verifying the benefits of SAM+PPO. In brief, we have i) additional comparison with a robust RL baseline called RARL for emphasizing the performance gains of SAM+PPO, ii) SAM with other policy-based methods, i.e., TRPO, for highlighting the applicability of SAM to other policy-based methods, iii) SAM+PPO in other environments, for confirming wide applicability of SAM+PPO in various settings, iv) Understanding how SAM+PPO enhances the robustness, by describing how SAM achieves better robustness across various factors than other related works.

---

> ### Author Response · Authors · 2024-11-24
> **Official Comment by Authors**
>
> **Weakness 2-1: Further ablations (additional comparison with RARL)**
>
> - We have included comparisons with additional baseline RARL (Robust Adversarial Reinforcement Learning) in our main experiments (in the submitted version, we add RARL only in complexity comparison, but we added all robustness experimental results in the revised paper).
> - In the revised paper, we added RARL results in Section 5, Experimental Results. We confirm that SAM+PPO consistently outperforms RARL in almost all robustness experiments, i.e., a wider range of robustness in friction-mass joint perturbation in Figure 6, a better robustness against reward perturbations in Table 2). RARL shows the best performance in the action robustness of the Hopper-v3 case (in Table 1), but it does not change the superiority of SAM+PPO in many cases.
> - We take the ‘action robustness’ part of Table 1 of the main paper, as follows:
>
> | Perturbation | Metric | HalfCheetah-v3 |  | Hopper-v3 |  | Walker2d-v3 |  |
> | --- | --- | --- | --- | --- | --- | --- | --- |
> |  |  | Nominal | Perturbed | Nominal | Perturbed | Nominal | Perturbed |
> | Action Noise σ = 0.2 | PPO | 4758 | 1469(−3289) | 3217 | 1467(−1750) | 4883 | 607(−4276) |
> |  | RNAC | 5484 | 2014(−3470) | 3445 | 1321(−2124) | 4147 | 652(−3495) |
> |  | RARL | 4996 | 3412(−1584) | 2819 | 1645(−1174) | 4020 | 764(−3256) |
> |  | SAM+PPO | **6523** | **4949(−1574)** | **3766** | **2312(−1454)** | **5129** | **2033(−3096)** |
>
> **Weakness 2-2: Further ablations (TRPO with SAM)**
>
> - We have considered TRPO as a new baseline and demonstrated how well SAM works in conjunction with TRPO.
> - Specifically, we extended our study by integrating SAM with TRPO, resulting in SAM+TRPO. TRPO uses trust region optimization to ensure stable policy updates by directly constraining the KL divergence between the old and new policies. As detailed in Appendix D, SAM+TRPO outperforms standard TRPO in several environments, indicating that the benefits of SAM are not limited to PPO. The results show that SAM enhances the robustness of TRPO, suggesting that promoting flatness in the loss landscape is beneficial across different optimization frameworks.
> | Perturbation | Metric | HalfCheetah-v3 |  | Hopper-v3 |  | Walker2d-v3 |  |
> | --- | --- | --- | --- | --- | --- | --- | --- |
> |  |  | Nominal | Perturbed | Nominal | Perturbed | Nominal | Perturbed |
> | Action Noise σ = 0.2 | TRPO | 4805 | 1502 (−3303) | 3118 | 1452 (−1666) | 4975 | 603 (−4372) |
> |  | SAM+TRPO | 5502 | 3975 (−1527) | 3547 | 2313 (−1234) | 5097 | 2052 (−3045) |
> | Mass Scale Factor 1.2 | TRPO | 4837 | 3865 (−972) | 3215 | 1556 (−1659) | 4957 | 782 (−4175) |
> |  | SAM+TRPO | 5562 | 5210 (−352) | 3499 | 3508 (+9) | 5205 | 5284 (+79) |
> | Friction Coefficient 0.88 | TRPO | 4723 | 4774 (+51) | 3075 | 1580 (−1495) | 4996 | 4756 (−240) |
> |  | SAM+TRPO | 5562 | 5539 (−23) | 3498 | 2728 (−770) | 5073 | 5134 (+61) |
>
> **Weakness 2-3: Further ablations (SAM+PPO in other environments)**
>
> - Furthermore, we conducted additional experiments on environments provided by Open-AI Gym, including environments with discrete action space, such as CartPole and LunarLander.
> - The new experimental results are included in Appendix D. We discuss how SAM+PPO performs in these settings and compare it with the provided baselines to show how SAM+PPO performs in settings with different action dynamics (we use action noise with $\sigma = 0.2$).
>
> | Algorithm | CartPole-v1 |  | LunaLander-v2 |  |
> | --- | --- | --- | --- | --- |
> |  | Nominal | Perturbed | Nominal | Perturbed |
> | PPO | 500 | 464(-36) | 200 | 175(-25) |
> | SAM+PPO | 500 | **481(-19)** | **200** | **188(-12)** |
> - Also, for the noisy reward cases, SAM+PPO shows the gains (we use reward noise with $\sigma = 0.1$; the noise is added in the training as done in the main experiment).
>
> | Algorithm | CartPole-v1 |  | LunaLander-v2 |  |
> | --- | --- | --- | --- | --- |
> |  | Nominal | Perturbed | Nominal | Perturbed |
> | PPO | 500 | 432(-68) | 200 | 165(-35) |
> | SAM+PPO | **500** | **458(-42)** | **200** | **182(-18)** |
> - From the theoretical viewpoint, we point out that SAM is a model-agnostic optimization function that can be applied to any gradient-based algorithm, including other policy gradient (PG) and actor-critic (AC) methods like TRPO, A2C, and SAC.
> - In our revised manuscript, we have added these results in Appendix D.1.

---

> ### Author Response · Authors · 2024-11-24
> **Official Comment by Authors**
>
> **Weakness 2-4: Understanding how SAM+PPO enhances the robustness**
>
> - The key essence of SAM to improve robustness is the indirect perturbations of actions, transitions, and rewards by perturbing model parameters. Thus, SAM can experience as various as actions that might be found by perturbing policy parameters. Therefore, SAM+PPO can be robust against any worst cases among the variations of actions, transitions, and rewards, leading to the generally well-generalized performance against the variations of diverse factors of environments.
> - In contrast, other action-robust RL methods, including RNAC and RARL, explicitly perturb the transition probabilities to experience the worst case. Specifically, RNAC considers the uncertainties in the transition probability of the environment and does robust polity optimization under the worst-case expected return over the uncertainty set. RARL adds adversarial perturbations to actions, maximizing its expected return under the exposure of adversarial perturbations. Therefore, these algorithms are strongly tailored to be robust against transition probability or actions but do not aim to achieve broad robustness across possible environmental variations. That is why SAM+PPO generally outperforms other robust RL methods across various perturbed settings.
>
> **Question 1: Clarification on the Perturbation Domain $\rho$ in Eq. 8**
>
> - In brief, we have added the pseudocode of SAM+PPO in Appendix C to describe the training steps fully. In the algorithm, $\rho$ is the radius of parameter perturbations $\epsilon$, where the loss values within the radius are minimized (referring to Min-Max problem of Eq. 8). **In training, $\rho$  is a kind of hyperparameter of SAM; thus the value is not given to the agent.**
> - When briefly explaining the steps in the pseudocode in Appendix C:
>     - Computing the base gradient $g_{\theta}=\nabla_{\theta}\mathcal{L}(\theta)$ at the parameter $\theta$
>     - Calculating the worst-case perturbation $\epsilon^* = \rho g_{\theta}/||g_{\theta}||$
>     - Performing an additional forward pass to obtain the loss computed at the perturbed parameter: $\mathcal{L}(\theta+\epsilon^*)$
>     - Computing the gradient $g_{\theta^*}=\nabla_{\theta*}\mathcal{L}(\theta^*)$, where $\theta^*=\theta+\epsilon^*$ (at the perturbed parameter)
>     - Updating the policy parameters using the gradient of the perturbed loss, i.e., $\theta \leftarrow \theta - \alpha g_{\theta^*}$, where $\alpha$ is the learning rate.
> - We hope this clarifies the details of the training steps of our algorithm.
>
> **Question 4: Proof of Proposition 1 needs a further revision**
>
> - We have revised the proof of Proposition 1 to ensure that the policy is consistently defined as a probability distribution throughout the proof.
> - As following the standard formulations of policies in PPO and TRPO, we represent the policy $\pi_{\theta}(a|s)$ as the Gaussian distribution with mean action $\mu_{\theta}(s)$ and fixed covariance matrix  $\Sigma$, i.e., $\pi_{\theta}(a|s) = \mathcal{N}(a; \mu_{\theta}(s), \Sigma)$. This approach is a standard form to handle continuous action space with probability distribution of policies. As a special note, we newly assume the Lipschitz continuity of reward, which is widely used in the related analysis of the robust RL.
>
> We truly appreciate your time for reviewing our work. We hope our response clarifies the raised concerns.

---

> > ### Comment · Reviewer_uQqo · 2024-11-25
> > **Thanks for the detailed response**
> >
> > thanks the authors for the time and effort preparing the response.
> > I am happy to raise the score.

---

> > > ### Author Response · Authors · 2024-11-25
> > >
> > > Thank you for your encouraging feedback and for taking the time to review our response. We appreciate your support and consideration.

---

### Official Review · Reviewer_RSs8 · 2024-11-01

**Soundness:** 3
**Presentation:** 2
**Contribution:** 3
**Rating:** 8
**Confidence:** 3

**Summary:**

The paper presents a study on using sharpness-aware regularization to obtain robust reinforcement learning policies. Drawing a theoretical connection between flatness in the reward, action and parameter space to action-robust RL, the authors present both a theoretical justification and experiments to show that the proposed method achieves good robustness properties.

**Strengths:**

The authors propose a simple yet intuitive approach for robust RL. I was somewhat surprised that this combination has apparently not been tried in the literature, but a brief literature survey has not brought up any similar algorithms. I actually think the authors are somewhat underselling their contributions here! While SAM has been used to train PPO before, the authors appropriately cite prior work here, previous papers have not drawn any connections to robust RL at all and the authors should feel entitled to proudly claim this connection as their connection! They do not merely provide theoretical backing, as far as I can tell, they make a connection that was wholly absent in cited work.

The theoretical statements are mostly correct as far as I can tell. See questions below however.

**Weaknesses:**

The main problem with the paper as it stands are writing problems and baseline comparisons.

Especially the beginning of the paper, abstract and introduction, suffer from very frequent grammar mistakes which make the paper much harder to read. I strongly encourage the author to revise the paper wrt to the writing.

In definition 1, I'm unsure if $\epsilon$ is added to the policy, parameters or action? From the proof it seems this is a parameter perturbation, this should be stated directly. I think adding parentheses in the equation would already make this much clearer, as we have two nested subscripts here.
In addition, the state is sampled from the policy, which seems strange?

As the theoretical statement depends on the Jacobian of the policy network, which is not bounded anywhere, I'm slightly skeptical that the theoretical results are sufficient to practically guarantee robust RL. Does the SAM objective guarantee or incentivize a flat Jacobian?

Given the surprisingly (?) bad results of RNAC - it barely seems to outperform PPO - I think it would be appropriate to apply SAM+PPO in the same environments as used in the RNAC paper. As far as I can tell, the code is available, so this should be feasible within the rebuttal timeline? If not, I will not hold this against the authors. I think it is important to verify that used examples are not cherry-picked to make the presented algorithm look stronger. This is the higher priority comment in terms of baseline comparisons.

I would encourage the authors to present some additional baselines. I acknowledge that more baselines is a somewhat lazy comment. However, given that there are several different formulations of robust RL, I believe it would be helpful to pick a variety of environments and algorithms presented with different robust formulations for comparison to understand how well the algorithm does in comparison to others. This doesn't have to be many or complex environments, just a larger variety of formalisms. This is a soft concern and not a large barrier to acceptance for me.
Both safe-control-gym [1] and Safety Gymnasium [2] provide a variety of tasks and implemented baselines to speed up experimentation.

[1] https://github.com/utiasDSL/safe-control-gym
[2] https://github.com/PKU-Alignment/safety-gymnasium

**Questions:**

Is there a specific advantage to using PPO with SAM, or could any PG or even AC algorithm be used? It might be that the clipping approximation to the trust region synergizes well with the SAM objective? I think this is an optional extension to the paper.

---

> ### Author Response · Authors · 2024-11-24
> **Official Comment by Authors**
>
> We thank the reviewer for their thoughtful feedback and valuable suggestions. As detailed below, we have addressed the weaknesses and questions raised.
>
> **Weakness 1: Correction mistakes in the Abstract and Introduction**
>
> - We apologize for any difficulties caused by grammatical errors in the abstract and introduction. Also, we greatly appreciate your careful suggestions. In the revised manuscript, we have thoroughly revised the entire paper and corrected grammatical errors, enhancing overall clarity.
>
> **Weakness 2: Clarification of Definition 1**
>
> - We have revised the notation to be clearer. We have revised Definition 1 to explicitly state that $\epsilon$ is the perturbation added to the policy parameter $\theta$.
>
> - Specifically, we have modified the expectation notation from
> $\mathbb{E}$$s,a\sim\pi_{\theta^*+\epsilon}$ to  $\mathbb{E}$${s \sim p, a \sim \pi_{\theta^* + \epsilon}(a|s)}$
>
> - Specifically, we have revised to indicate that the state $s$ is sampled from the transition probability distribution $s\sim p$, and the action $a$ is sampled from the perturbed policy $a\sim\pi_{\theta^*+\epsilon}(a|s)$.
>
> **Weakness 3: Unbounded Jacobian and Practical Guarantees**
>
> - In practice, it is hard to guarantee bounded Jacobian of deep neural networks. However, popular techniques such as weight regularization, gradient clipping, and bounded activation functions help control the magnitude of the network's weights and gradients, effectively constraining the Jacobian.
> - When considering the SAM’s objective, SAM promotes convergence to flatter regions in the loss landscape by considering parameter perturbations during optimization. This process inherently discourages sharp changes in the loss with respect to parameter changes, which is related to the second derivative (Hessian) and indirectly to the Jacobian. Also, when SAM finds loss minima, SAM suppresses the norm of gradients, making the Jacobian smoother and more stable around the minima. It is analogous to the main objective of SAM to find smoother loss surfaces around minima.
> - Our empirical results demonstrate that SAM+PPO leads to policies that are more robust to perturbations, supporting the practical relevance of our theoretical findings.
> - We have added the discussion for the bounds of Jacobian in the right after the proof of our Proposition in the revised paper.
>
> **Weakness 4: Baseline comparisons and the tests in the RNAC environments**
>
> - We appreciate your careful comments on the baseline comparisons, along with suggestions for the open-source code of RNAC.
> - We emphasize that we have done experiments for RNAC with the shared code, and the hyperparameters of RNAC are the same as those used in the original paper. Therefore, we are quite sure that the RNAC experiments in our paper do not have any technical mistakes or intentional picking of worse cases.
> - To figure out the reason behind the not well-performing results of RNAC, we want to point out that the perturbation settings in the RNAC paper are not as challenging as we did in our paper. Specifically, the RNAC paper’s robustness evaluation is conducted with perturbation of changing stiffness of the actuator joint (e.g., leg joint stiffness), which is generally easier for agents to handle. It directly affects the actuator joint where the action is applied, making it easy to predict and adjust to the changes. In contrast, we evaluated the robustness of the change in mass and ground friction, which introduce indirect and widespread effects on the dynamics. The agent is required to infer and adapt to more complex interactions in the environment. Also, we widely tested robustness against action and reward, which is not covered in the RNAC paper. We conjecture that RNAC is less effective in handling these varieties of perturbations than SAM+PPO.
> - Also, even in the original RNAC paper, it does not outperform PPO but shows modest gains over PPO. This suggests that the performance differences between RNAC and PPO are consistent across the RNAC paper and ours.
> - In conclusion, we want to emphasize that our empirical comparisons are not cherry-picking, and our simulations of RNAC are reliable. Furthermore, it confirms that RNAC is less robust in various factors of environmental changes.

---

> ### Author Response · Authors · 2024-11-24
> **Official Comment by Authors**
>
> **Weakness 5 & Question 1: Tests in additional environments and baselines**
>
> - **Additional baselines:** We have included comparisons with additional baseline RARL (Robust Adversarial Reinforcement Learning) in our main experiments (in the submitted version, we added RARL only in complexity comparison, but we added all robustness experimental results in the revised paper).
> - In the revised paper, we added RARL results in Section 5. Experimental Results. We confirm that SAM+PPO consistently outperforms RARL in almost all robustness experiments, i.e., a wider range of robustness in friction-mass joint perturbation in Fig. 6, a better robustness against reward perturbations in Table 2). RARL shows the best performance in the action robustness of the Hopper case (in Table 1), but it does not change the superiority of SAM+PPO in many cases.
> - We take the ‘action robustness’ part of Table 1 of the main paper, as follows:
>
> | Perturbation | Metric | HalfCheetah-v3 |  | Hopper-v3 |  | Walker2d-v3 |  |
> | --- | --- | --- | --- | --- | --- | --- | --- |
> |  |  | Nominal | Perturbed | Nominal | Perturbed | Nominal | Perturbed |
> | Action Noise σ = 0.2 | PPO | 4758 | 1469(−3289) | 3217 | 1467(−1750) | 4883 | 607(−4276) |
> |  | RNAC | 5484 | 2014(−3470) | 3445 | 1321(−2124) | 4147 | 652(−3495) |
> |  | RARL | 4996 | 3412(−1584) | 2819 | **1645(−1174)** | 4020 | 764(−3256) |
> |  | SAM+PPO | **6523** | **4949(−1574)** | **3766** | 2312(−1454) | **5129** | **2033(−3096)** |
>
> - To answer Question 1, we have considered TRPO as a new baseline and demonstrated how well SAM works in conjunction with TRPO.
> - Specifically, we extended our study by integrating SAM with TRPO, resulting in SAM+TRPO. TRPO uses trust region optimization to ensure stable policy updates by directly constraining the KL divergence between the old and new policies. As detailed in Appendix D, SAM+TRPO outperforms standard TRPO in several environments, indicating that the benefits of SAM are not limited to PPO. The results show that SAM enhances the robustness of TRPO, suggesting that promoting flatness in the loss landscape is beneficial across different optimization frameworks.
>
> | Perturbation | Metric | HalfCheetah-v3 |  | Hopper-v3 |  | Walker2d-v3 |  |
> | --- | --- | --- | --- | --- | --- | --- | --- |
> |  |  | Nominal | Perturbed | Nominal | Perturbed | Nominal | Perturbed |
> | Action Noise σ = 0.2 | TRPO | 4805 | 1502 (−3303) | 3118 | 1452 (−1666) | 4975 | 603 (−4372) |
> |  | SAM+TRPO | **5502** | **3975 (−1527)** | **3547** | **2313 (−1234)** | **5097** | **2052 (−3045)** |
> | Mass Scale Factor 1.2 | TRPO | 4837 | 3865 (−972) | 3215 | 1556 (−1659) | 4957 | 782 (−4175) |
> |  | SAM+TRPO | **5562** | **5210 (−352)** | **3499** | **3508 (+9)** | **5205** | **5284 (+79)** |
> | Friction Coefficient 0.88 | TRPO | 4723 | **4774 (+51)** | 3075 | 1580 (−1495) | 4996 | 4756 (−240) |
> |  | SAM+TRPO | **5562** | 5539 (−23) | **3498** | **2728 (−770)** | **5073** | **5134 (+61)** |
> - **Additional environments:** Furthermore, we conducted additional experiments on environments provided by Open-AI Gym, including environments with discrete action space, such as CartPole and LunarLander.
> - The new experimental results are included in Appendix D. We discuss how SAM+PPO performs in these settings and compare it with the provided baselines to show how SAM+PPO performs in settings with different action dynamics (we use action noise with $\sigma = 0.2$).
>
> | Algorithm | CartPole-v1 |  | LunaLander-v2 |  |
> | --- | --- | --- | --- | --- |
> |  | Nominal | Perturbed | Nominal | Perturbed |
> | PPO | **500** | 464(-36) | **200** | 175(-25) |
> | SAM+PPO | **500** | **481(-19)** | **200** | **188(-12)** |
> - Also, for the noisy reward cases, SAM+PPO shows the gains (we use reward noise with $\sigma = 0.1$; the noise is added in the training as done in the main experiment).
>
> | Algorithm | CartPole-v1 |  | LunaLander-v2 |  |
> | --- | --- | --- | --- | --- |
> |  | Nominal | Noisy | Nominal | Noisy |
> | PPO | **500** | 432(-68) | **200** | 165(-35) |
> | SAM+PPO | **500** | **458(-42)** | **200** | **182(-18)** |
> - From the theoretical viewpoint, we point out that SAM is a model-agnostic optimization function that can be applied to any gradient-based algorithm, including other policy gradient (PG) and actor-critic (AC) methods like TRPO, A2C, and SAC.
> - In our revised manuscript, we have added these results in Appendix D.1.

---

> ### Author Response · Authors · 2024-11-24
> **Official Comment by Authors**
>
> - To answer Question 1, we have considered TRPO as a new baseline and demonstrated how well SAM works in conjunction with TRPO.
> - Specifically, we extended our study by integrating SAM with TRPO, resulting in SAM+TRPO. TRPO uses trust region optimization to ensure stable policy updates by directly constraining the KL divergence between the old and new policies. As detailed in Appendix D, SAM+TRPO outperforms standard TRPO in several environments, indicating that the benefits of SAM are not limited to PPO. The results show that SAM enhances the robustness of TRPO, suggesting that promoting flatness in the loss landscape is beneficial across different optimization frameworks.
>
> | Perturbation | Metric | HalfCheetah-v3 |  | Hopper-v3 |  | Walker2d-v3 |  |
> | --- | --- | --- | --- | --- | --- | --- | --- |
> |  |  | Nominal | Perturbed | Nominal | Perturbed | Nominal | Perturbed |
> | Action Noise σ = 0.2 | TRPO | 4805 | 1502 (−3303) | 3118 | 1452 (−1666) | 4975 | 603 (−4372) |
> |  | SAM+TRPO | **5502** | **3975 (−1527)** | **3547** | **2313 (−1234)** | **5097** | **2052 (−3045)** |
> | Mass Scale Factor 1.2 | TRPO | 4837 | 3865 (−972) | 3215 | 1556 (−1659) | 4957 | 782 (−4175) |
> |  | SAM+TRPO | **5562** | **5210 (−352)** | **3499** | **3508 (+9)** | **5205** | **5284 (+79)** |
> | Friction Coefficient 0.88 | TRPO | 4723 | **4774 (+51)** | 3075 | 1580 (−1495) | 4996 | 4756 (−240) |
> |  | SAM+TRPO | **5562** | 5539 (−23) | **3498** | **2728 (−770)** | **5073** | **5134 (+61)** |
> - While we focused on integrating TRPO and additional OpenAI Gym environments due to the limited rebuttal period, we highly acknowledge the value of safe-control-gym and Safety Gymnasium, which offers a rich set of environments and baseline implementations for safe and robust RL. We plan to incorporate environments and baselines from these resources in future research to further evaluate our method in safety-critical settings and against a broader range of robust RL formulations.
> - **Synergy with PPO's Clipping Mechanism:** PPO's clipping mechanism serves as an approximation to trust region optimization, preventing large updates that could destabilize training. This clipping synergizes well with SAM's objective of seeking flat minima by limiting the parameter updates to a stable region. PPO approximates trust region optimization through clipping, ensuring that the updated policy does not deviate excessively from the old policy. SAM complements this by seeking parameter regions where such deviations are less sensitive to perturbations.
>
> We truly appreciate your time for reviewing our work. We hope our response clarifies the raised concerns.

---

> > ### Comment · Reviewer_RSs8 · 2024-11-24
> > **Thanks**
> >
> > Thanks for the very thorough reply. I’m happy to recommend acceptance.

---

> > > ### Author Response · Authors · 2024-11-24
> > >
> > > Thank you for your positive feedback and recommendation. I appreciate your time and effort in reviewing my work.

---

### Official Review · Reviewer_jacH · 2024-11-03

**Soundness:** 3
**Presentation:** 2
**Contribution:** 2
**Rating:** 8
**Confidence:** 3

**Summary:**

The paper investigates the relationship between flat reward maxima in policy parameter space and the robustness of reinforcement learning (RL) agents. It claims that flatter reward maxima lead to more robust policies, particularly against action perturbations. The paper presents a theoretical proposition linking flat reward to action robustness and supports this claim through empirical experiments in MuJoCo environments (e.g., Hopper-v3, Walker2d-v3, HalfCheetah-v3). The authors demonstrate that an RL algorithm enhanced with Sharpness-Aware Minimization (SAM), called SAM+PPO, consistently outperforms standard PPO and a recent robust RL baseline (RNAC) in various robustness tests, including action noise, transition probability changes, and reward function variations. The paper also provides visualizations and quantitative measurements of reward surfaces, further confirming the link between flatness and robustness.

**Strengths:**

- This paper provides a formal link between flat reward surfaces and robustness in policy space. Proposition 1 establishes a clear theoretical foundation for the paper's main claim.
- The authors comprehensively test SAM+PPO across multiple challenging environments and scenarios, including noisy actions and varying transition probabilities, to demonstrate robustness.
- The authors compare SAM+PPO with RNAC, PPO, and RARL, which shows both performance and computational efficiency, which strengthens their findings.
- The use of reward surface visualizations and flatness metrics strengthens the paper's argument by providing visual and quantitative evidence for the flatness achieved by SAM+PPO.

**Weaknesses:**

- While SAM is shown to be effective, the paper lacks a discussion of its potential limitations, such as computational overhead or sensitivity to hyperparameter tuning.
- The justification for reward noise being added during training for reward function robustness evaluation could be clearer: The paper mentions this difference in methodology but could expand on why this is necessary for a valid evaluation.
- I don't know if the preliminary experiment is best placed in the introduction, it feels a bit out of place for me.
- typos 234 "objeective", 249 " funciton"

**Questions:**

- Do you have an intuition on why SAM doesn't perform better on Walker2d-v3 for high friction factor?
- Have you tested SAM+PPO on non-MuJoCo environments to assess robustness in discrete action spaces or varying reward structures?

---

> ### Author Response · Authors · 2024-11-24
>
> We thank the reviewer for their thoughtful feedback and valuable suggestions. As detailed below, we have addressed the weaknesses and questions raised.
>
> **Weakness 1-1: Potential limitations (computational overhead)**
> - SAM+PPO requires additional computations in the optimization process. When describing in detail, let us elaborate on the steps to optimize the cost function in Eq. 8, which is the SAM+PPO’s objective function.
> - Integrating SAM into PPO requires modifying the standard optimization steps to account for the perturbation ϵ. This involves:
>     - Computing the base gradient $g_{\theta}=\nabla_{\theta}\mathcal{L}(\theta)$ at the parameter $\theta$
>     - Calculating the worst-case perturbation $\epsilon^* = \rho g_{\theta}/||g_{\theta}||$
>     - Performing an additional forward pass to obtain the loss computed at the perturbed parameter: $\mathcal{L}(\theta+\epsilon^*)$
>     - Computing the gradient $g_{\theta^*}=\nabla_{\theta*}\mathcal{L}(\theta^*)$, where $\theta^*=\theta+\epsilon^*$ (at the perturbed parameter)
>     - Updating the policy parameters using the gradient of the perturbed loss, i.e., $\theta \leftarrow \theta - \alpha g_{\theta^*}$, where $\alpha$ is the learning rate.
> - This process introduces one additional gradient computation in each optimization step, doubling the required number of forward and backward computations compared to the standard PPO.
> - In big $\mathcal{O}$ notation, the per-iteration computational complexity increases from $\mathcal{O}(N)$ for PPO to $\mathcal{O}(2N)$ for SAM+PPO, where $N$ is the number of parameters.
> - In actual training, we have additionally measured the training time per optimization step (or iteration) of PPO and SAM+PPO. As shown in the following table, SAM+PPO incurs approximately 1.5 times larger time per model update for all experiments, which seems a reasonable trade-off considering the robustness gains.
>
> | Algorithm | HalfhCheetah | Hopper | Walker |
> | --- | --- | --- | --- |
> |  | Train | Train | Train |
> | PPO | 1.22 | 0.13 | 0.2 |
> | SAM+PPO | 1.5(x1.83) | 0.23(×1.76)  | 0.24(×1.20)  |
>
> - In the revised manuscript, we have added the aforementioned step-by-step descriptions of SAM+PPO’s optimization at Algorithm 1 in Appendix C.
> - Also, in the revised manuscript, we have added the training time per iteration at Table C.1 in Appendix C.
>
> **Weakness 1-2: Potential limitations (sensitivity of hyperparameters)**
> - A newly introduced hyperparameter for SAM+PPO beyond PPO is $\rho$, i.e., the radius of the perturbations. The value of ρ directly affects the degree of flatness sought in the reward landscape. A larger ρ encourages the optimizer to find flatter maxima, potentially enhancing robustness but at the risk of training instability (too wide flat minima are hard to find). Conversely, a smaller ρ may result in less robustness gain. Selecting an appropriate ρ is crucial not only in our RL cases but also in other learning tasks.
> - We have additionally provided the sensitivity of the performance of Hopper by changing $\rho$:
>
> | Algorithm | Nominal | Action Noise  σ = 0.05 | Action Noise  σ = 0.1 | Action Noise  σ = 0.15 | Action Noise  σ = 0.2 | Action Noise  σ = 0.25 |
> | --- | --- | --- | --- | --- | --- | --- |
> | PPO | 3217 | 2083 | 1792 | 1577 | 1467 | 1284 |
> | SAM+PPO($\rho$ 0.001) | 3329 | 2214 | 1619 | 1515 | 1291 | 1085 |
> | SAM+PPO($\rho$ 0.005) | 3294 | 2795 | 1853 | 1305 | 1077 | 972 |
> | SAM+PPO($\rho$ 0.01) | **3766** | **3732** | **3589** | **3123** | **2312** | **1917** |
> | SAM+PPO($\rho$ 0.05) | 2191 | 2019 | 2038 | 1920 | 1732 | 1782 |
> | SAM+PPO($\rho$ 0.1) | 2735 | 2716 | 2306 | 2077 | 1781 | 1739 |
>
> - We have selected $\rho=0.01$ for the optimized hyperparameter. It is hard to say that SAM works well in a broad range of $\rho$ values, but it is quite common sensitivity of the perturbation radius observed in other SAM-based experiments for visual classifications.
> - In addition, we found that the choice of $\rho=0.008$, which is similar to the choice for Hopper, is shown to be optimal for HalfCheetah and Walker2d. This means that a similar level of perturbation radius is required for the three environments.
> - For other hyperparameters of PPO, the application of SAM on PPO does not inherently change the role or sensitivity of PPO's original hyperparameters. Thus, we have simply utilized the same hyperparameters for PPO and SAM+PPO, relieving the efforts to tune the hyperparameters.
> - We have added the sensitivity results of $\rho$ in Appendix C.3.

---

> ### Author Response · Authors · 2024-11-24
>
> **Weakness 2: Justification for adding reward noise during training**
> - Let us anticipate a case where the training is done without reward noise, but the evaluation is done with reward noise. Adding reward noise during evaluation would NOT affect the agent's behavior because the policy decisions are made before observing the rewards. Thus, the noisy version of rewards would be observed in testing, which does not provide meaningful insights into the policy’s robustness. That is why we adopt noisy rewards during the training phase.
> - Also, in practical scenarios, it is important to guarantee the robustness of agents against noisy reward observations.  Agents often operate in real-world environments where the reward signals during training are noisy due to measurement errors or uncertainties. Training agents under such conditions and evaluating them in the nominal environment allows us to assess their ability to learn effective policies despite these challenges.
>
> **Weakness 3: About the preliminary experiments in the Introduction**
>
> - We appreciate your careful suggestion about the preliminary experiments in the Introduction. First, we intend to provide a conceptual understanding of ‘action robustness’ by showing the experiments. When navigating with a risk of erroneous actions, an agent has to keep the margin space to the obstacles. Therefore, we believe that the 2D maze environment clearly shows how the agent behaves to guarantee the robustness of the action by avoiding the risky narrow path.
> - We carefully considered its placement and aimed to ensure that it enhances the reader's understanding of the motivation behind our research.
> To address your concern about the experiment feeling out of place, we have revised the introduction to better integrate the preliminary experiment:
> We have refined the transition into the preliminary experiment to ensure it flows naturally from the background and motivation. We clarified the purpose of the preliminary experiment, emphasizing how it illustrates the necessity for our proposed method
> - Also, we have redesigned Figure 1 by adding the mini-figure that has increased the zoom level, to make key features more visible, illustrating the essence of the preliminary experiments.
>
> **Question 1: Discussions on Walker2d-v3 with high friction**
>
> - It is challenging to fully understand the empirical results, particularly in the high friction regime. However, we have tried our best to explain the results as follows:
> - First of all, compared with HalfCheetah-v3 and Hopper, the Walker2d-v3 environment is particularly sensitive to changes in friction due to the complex dynamics of balancing and coordinating two legs.
> - Second, the increasing friction from low to high can be understood as the dramatic changes in environments. The agent of Walker2d-v3 has its center of mass at the upper side of its body compared with HalfCheetah-v3 and Hopper-v3. Thus, the high friction makes the agent easily fall over.
> - Third, SAM does not directly perturb actions but indirectly perturbs actions via perturbing the policy parameters. So, SAM can experience as various as actions that might be found by perturbing policy parameters. In contrast, other action robust RL explicitly perturbs the action to experience the worst case.
> - Based on the aspects above, we conjecture that SAM+PPO focuses less on experiencing high friction cases, which are extreme perturbations in environments.
> - However, we want to point out that SAM+PPO can widely experience actions, transitional probability, and reward shifting by “indirectly” perturbing the policy parameters. As empirical evidence, SAM+PPO outperforms PPO in the various factors of perturbations, but RNAC or RARL shows comparably limited gains over PPO (referring to Figure 6 and Table 1 in the main paper).

---

> ### Author Response · Authors · 2024-11-24
>
> **Question 2: Testing on Non-MuJoCo Environments**
>
> - We have conducted additional experiments on non-MuJoCo environments provided by Open-AI Gym, including environments with discrete action space, such as CartPole-v1 and LunarLander-v2.
> - For CartPole-v1 and LunaLander-v2, SAM+PPO shows consistent gains over PPO for the action robustness experiments (we use action noise with $\sigma = 0.2$).
>
> | Algorithm | CartPole-v1 |  | LunaLander-v2 |  |
> | --- | --- | --- | --- | --- |
> |  | Nominal | Perturbed | Nominal | Perturbed |
> | PPO | 500 | 464(-36) | 200 | 175(-25) |
> | SAM+PPO | **500** | **481(-19)** | **200** | **188(-12)** |
> - As shown in the upper table, we confirm that SAM+PPO consistently outperforms PPO by taking advantage of flat rewards. Also, these results support our claim that the flat reward promotes the robustness of RL even in non-MuJoCo environments.
> - Also, for the noisy reward cases, SAM+PPO shows the gains (we use reward noise with $\sigma = 0.1$; the noise is added in the training as done in the main experiment).
>
> | Algorithm | CartPole-v1 |  | LunaLander-v2 |  |
> | --- | --- | --- | --- | --- |
> |  | Nominal | Noisy | Nominal | Noisy |
> | PPO | 500 | 432(-68) | 200 | 165(-35) |
> | SAM+PPO | **500** | **458(-42)** | **200** | **182(-18)** |
> - In our revised manuscript, we have added these results in Appendix D.1.
>
> We truly appreciate your careful and thoughtful reviews. We hope our response clarifies your concerns.

---

> > ### Comment · Reviewer_jacH · 2024-11-25
> > **thank you**
> >
> > I would like to thank the authors for the very thorough and detailed reply. After going through it and reading other reviewers discussions, I’m happy to recommend acceptance.

---

> > > ### Author Response · Authors · 2024-11-26
> > >
> > > Thank you for your positive feedback and for recommending acceptance. We appreciate your thorough review, which helped us improve the paper.

---

### Author Response · Authors · 2024-11-24
**Overall Comments : Summary of revisions to the original paper**

We would like to express our sincere gratitude for your thoughtful and constructive feedback on our manuscript (highlighted in blue-colored font).

We have carefully considered all your comments and have made comprehensive revisions to address your concerns.

Here, we provide a detailed list of the changes made to the manuscript, organized by sections:

- **Abstract & Section 1. Introduction**
    - **Grammar correction** : We have thoroughly revised the entire paper, corrected grammatical errors, enhanced overall clarity.
    - **Preliminary experiment**
        - Refined the transition into the preliminary experiment to ensure it flows naturally from the background and motivation. We clarified the purpose of the preliminary experiment, emphasizing how it illustrates the necessity for our proposed method.
        - Redesigned Figure 1 by adding the mini-figure that has increased the zoom level, to make key features more visible, illustrating the essence of the preliminary experiments.
- **Section 3. Preliminaries**
- **Equation 1. clarification** :  $\mathbb{E}$${s, a \sim \pi}$ to $\mathbb{E}$${s \sim p, a \sim \pi}$
- **Details of Sharpness Aware Minimization**  : Provided in Appendix C.
- **Section 4. Linking Flat Reward to Action Robustness**
  - **Definition 1. clarification** :  $\mathbb{E}$${s, a \sim \pi_{\theta^* + \epsilon}}$ to $\mathbb{E}$${s \sim p, a \sim \pi_{\theta^* + \epsilon}(a|s)}$
- **Added Figure 2** : Visualization of Definitions 1, 2
    - **Proposition 1 clarification** : Revised the statement to understand the link between Definition 1, 2
    - **Remark 1.2 clarification** : ‘For changes of reward function’ → ’For reward function perturbations,. etc.
- **Section 5. Experimental Results**
    - **Extended evaluation :** Added results of RNAC and RARL across all evaluation
- **Appendix A. Proof of Proposition 1**
    - **Policy stated as probability distribution** : Revised the proof to ensure that the policy is consistently defined as a probability distribution throughout the proof
    - **Added Appendix A.2** : Discussion on the bounds of the Jacobian
- **Appendix C. Details and Analysis of SAM integrated with PPO**
    - **Detailed discussion** : Provided a detailed discussion on the integration of SAM with RL
- **Appendix D. Additional Experimental Results**
    - **SAM enhanced other RL algorithm(SAM+TRPO)** :  Added to validate the applicability and reliability of our SAM-enhanced method in a broader contex
    - **Discrete action environments provided by OpenAI Gym(Cartpole-v1, Lunarlander-v2)** : Added to assess the performance of SAM+PPO in settings with discrete action
    spaces and different reward structures.
- **Appendix E. Comparison with existing Robust RL algorithms**
    - **Reorganized content** : Moved the extra experiment from Section 5.6 in the original paper to Appendix E
    - **Ablation studies** : Provide ablation to understand how SAM enhanced RL bring benefits in comparison with other algorithms

---

### Meta-Review · Area_Chair_EFuD · 2024-12-21

**Metareview:**

The paper investigates the relationship between flat minima and robustness in RL, finding that flatter minima correspond to more robust policies. These theoretical claims are supported with empirical results, including a new variant of PPO that uses a sharpness aware minimizer.

The reviewers appreciated the novelty of the analysis in the paper, the clear theoretical results, and the strong empirical results of the proposed method. The reviewers also appreciated the use of visualizations and metrics to bolster claims relating flatness with robustness. The reviewers note that sharpness aware minimizers have been used in conjunction with PPO before, but appreciate that the authors clearly describe the relationship and contributions relative to this prior work. Finally, the reviewers appreciate the improved computational complexity and sample complexity of the proposed method.

The reviewers had some suggestions for clarifying the paper writing (grammar, statements of some of the theoretical results) and suggestions for additional baselines and visualizations.

Overall, this is a very well written paper about a novel perspective on robustness, and should be appreciated by many members of the ICLR community.

**Additional Comments On Reviewer Discussion:**

During a robust discussion period, the authors addressed concerns about computational complexity, hyperparameter sensitivity, ablations, and applicability to new tasks. The reviewers also revised the paper (e.g., definitions). One reviewer explicitly lauded the authors for revising the paper (during the rebuttal) in terms of experiments, theory, and presentation.

---

### Decision · Program_Chairs · 2025-01-22

Accept (Oral)